civil engineering

rockburst, roadway support, fracture evolution, soft structure, multi-length support technology

**Author for correspondence:**
Mingshi Gao
e-mail: cumt_gms@cumt.edu.cn

# Investigation of the evolution and control of fractures in surrounding rock under different pressure relief and support measures in mine roadways prone to rockburst events

Yongliang He[1,2], Mingshi Gao[1,2], Dong Xu[1,2] and Xin Yu[1,2]

[1]School of Mines, China University of Mining and Technology, Xuzhou, Jiangsu 221116, People's Republic of China
[2]State Key Laboratory of Coal Resource and Safe Mining, China University of Mining and Technology, Xuzhou, Jiangsu 221116, People's Republic of China

YH, 0000-0002-8314-790X

This paper studies the evolution and control of surrounding rock under different pressure relief support conditions in mine roadways in which rockburst events have occurred. The evolution of fractures in the surrounding rock was determined from borehole images obtained with a digital panoramic borehole camera, and the surface displacement due to the rockburst events in the mine roadway was measured. According to the existing problems of the original support system of the roadway, a new coupled support system to prevent rockburst events in mine roadways was proposed, resolving both the pressure relief and support of the roadway. Field measurements indicate that the effect on the roadway under the coupled method of pressure relief and support was more satisfactory than that under the original support system. With the coupled support method, the surface displacement of the roadway was approximately 0.6 m, fractures were distributed only in the soft structures and bolt anchorage areas, and the maximum depth of the fractures was 2.95 m. By contrast, under the original support system, fractures were distributed throughout the roadway surrounding rock, and the maximum depth of fractures was 6.75 m. This coupled roadway support technology of pressure relief and support effectively

maintains the stability of the rock surrounding the roadway and ensures the safety of the working face. The research results can provide a reference for damage prevention and support of mine roadways prone to rockburst events.

## 1. Introduction

Rockburst is the sudden rupture and ejection of coal and/or rock from the surface of a roadway or working face. This process instantaneously releases a large amount of energy, which is accompanied by a loud noise; rockburst may cause equipment and roadway damage, lead to gas and coal dust explosions and even result in casualties [1–5]. With the gradually increasing depth of coal mining, mine depths have exceeded 1000 m and reached 1500 m, and the intensity and frequency of rockbursts have increased significantly [6]. In China, the number of coal mines that have experienced rockburst increased from 142 in 2013 to 177 in 2017 [7]. Figure 1 shows the number and distribution of mines prone to rockburst operating in China in June 2019. From 2015 to 2020, partial rockburst accidents with serious casualties occurred in roadways across China, as shown in table 1. According to statistics, 85% of rockburst accidents occur in roadways [8].

There are many studies on controlling roadway deformation and fracture propagation through pressure relief and support measures. Pressure relief methods include deep hole pre-split blasting [9–11], roof directional hydraulic fracturing [12,13], coal seam blasting pressure relief [14–16], coal seam large-diameter borehole pressure relief, coal seam high-pressure water injection [17], floor blasting and floor cutting.

Kang and co-workers [18,19] found that coal and rock masses form the main bodies of roadways and bear dynamic and static loads, and that the application of rock bolts improve this bearing capacity. The theory of high pre-stress and strong support is proposed to meet the initial support requirements of the roadway and avoid roadway destruction. Ju and co-workers [20,21] suggested the equivalent section support principle for the internal stress arch structure and considered the characteristics of the full-section appearance of rockburst events in mine roadways. Gao and co-workers [22,23] proposed the strong–soft–strong structure theory of rockburst events in mine roadways and analysed their wave absorption and energy absorption characteristics, failure criterion and energy absorption effect. Pan and co-workers [24,25] described a flexible-coupling fast-yielding support based on pendulum wave theory, fast energy absorption and yielding support. Lü & Pan [26] studied the propagation and attenuation of stress waves in roadways, obtained the energy and stress criterion of stress waves in the surrounding rock, and proposed a rigid-flexible coupled support system. Pan *et al.* [27,28] proposed multi-scale source control to prevent and control deep coal and rock dynamic disasters and studied the theory and technology of dynamic and static load sources to prevent and control deep mine roadway rockbursts. Wu *et al.* [29] and Kang *et al.* [30] assessed the response of rock bolts under dynamic and static loads in coal mines under impact, and several series of high-strength rock bolts were developed with different rock bolt strengths, elongations and impact toughnesses. He and co-workers [31–35] developed constant resistance large deformation (CRLD) bolts and established a dynamic CRLD bolt model. An anchor cable that exhibits a high constant resistance and large deformation produces instantaneous deformation under dynamic loading; this cable has special mechanical properties and can maintain a constant resistance.

With the Changcun coal mine as the research site, this paper proposes a method for pressure relief and support coordination. The characteristics of roadway deformation and the fracture evolution in the surrounding rock are studied with a borehole camera to reveal the pressure relief and support coordination mechanism. The coupled technology of pressure relief and support is proposed to realize the safety and stability of mine roadways prone to rockbursts. This new technology design, guaranteeing sustainable mining production, also provides some practical reference for solving problems similar to rockburst in mine roadways.

## 2. General introduction to Changcun coal mine

### 2.1. Geologic conditions

The Changcun coal mine is located in Changcun town, Sanmenxia city, Henan Province, China. The Changcun coal mine is designed to adopt inclined shaft development. The direct roof of the roadway is sandstone with a thickness of 0.1–9.5 m. The sandstone [36] contains water, the lower part is loose and collapses easily, and the upper part is dense and hard. The main roof is mudstone [37] with a thickness of 35.5–44.5 m. The mudstone is thick but weathers and breaks easily. The direct bottom is

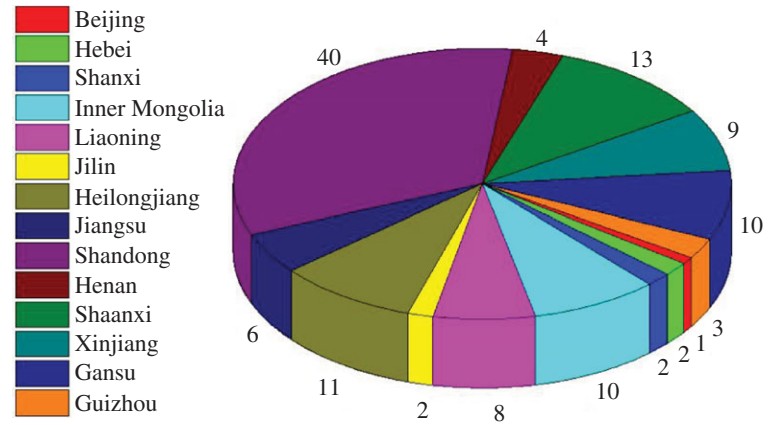

**Figure 1.** Number and distribution of mines prone to rockburst in China (June 2019).

**Table 1.** Partial rockburst events in mine roadways leading to accidents in China from 2015 to 2020.

| date | rockburst coal mine | rockburst situation |
| --- | --- | --- |
| 26 May 2015 | Liaoning Fuxin Mining Group, Aiyou coal mine | killed 4 coal miners |
| 22 Dec 2015 | Henan Dayou Energy Co., Ltd., Gengcun coal mine | roadway damage 160 m; killed 2 coal miners |
| 22 Jul 2016 | Heilongjiang Longmei Group, Dongbaowei coal mine | roadway damage 120 m; killed 2 coal miners |
| 15 Aug 2016 | Shandong energy Feicheng Mining Group, Liangbaosi Energy Co., Ltd. | killed 2 coal miners |
| 17 Jan 2017 | China Coal Group, Danshuigou Coal Industry Co., Ltd. | killed 10 coal miners |
| 11 Nov 2017 | Liaoning Shenyang Coking Coal Co., Ltd., Hongyang No. 3 coal mine | roadway damage 220 m; killed 10 coal miners |
| 20 Oct 2018 | Shandong Energy Group, Longyun Coal Industry Co., Ltd. | roadway damage 100 m; killed 21 coal miners |
| 9 Jun 2019 | Jilin Longjiabao Mining Co., Ltd. | roadway damage 220 m; killed 9 coal miners |
| 2 Aug 2019 | Kailuan (Group) Co., Ltd, Tangshan Mining Branch | killed 7 coal miners |
| 22 Feb 2020 | Shandong Xinjulong Energy Co., Ltd. | killed 4 coal miners |

carbonaceous mudstone with a thickness of 0.5–5.7 m; it is grey-black and massive, has a high carbon content and readily swells when in contact with water. The main floor is 27.2–32.8 m clay rock with argillaceous cementation, as shown in figure 2a. The ground elevation of the 21 170 working face is 497–549 m, and the working face elevation is −161.080 to −210.709 m. As shown in figure 2b, the upper part of the 21 170 working face is the 21 150 working face that has been mined, and the lower part is the 21 190 working face that has not been mined yet.

## 2.2. Rockburst conditions and roadway failure characteristics

The burial depth of the 21 170 working face exceeds 700 m, and the vertical stress exceeds 17.5 MPa. Due to the extensive concentration of production, the excavation interference is large and the mining intensity is gradually increasing, resulting in rockburst. During the mining and production processes, dozens of shock events have occurred, and the energy is mainly concentrated at approximately $10^4 - 10^6$ J. However, observations of the multiple rockburst events in the mine roadways suggest that rockburst at this site greatly damages the roadway support and disrupts the surrounding rock stability. After a

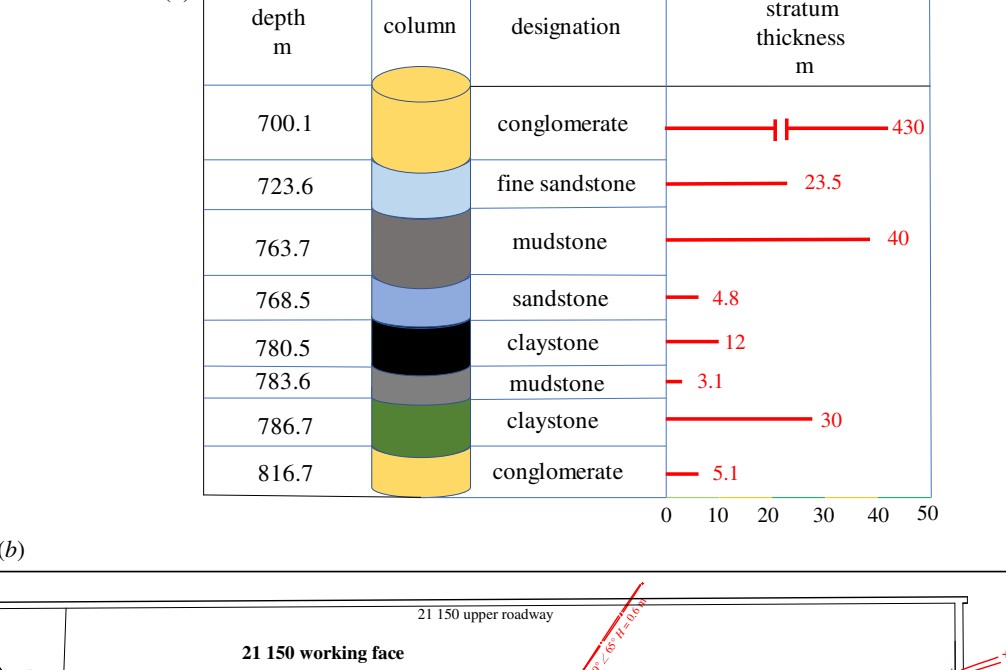

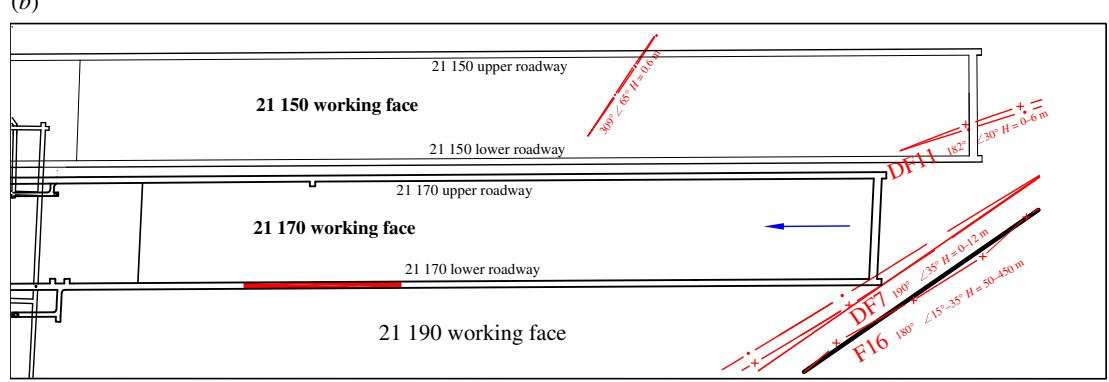

**Figure 2.** General situation of the Changcun coal mine. (*a*) Stratigraphy of the test roadway. (*b*) Layout of the test areas.

rockburst occurs, it causes varying degrees of damage to the roadway, such as damage to the roadway support, belt bulging and twisting, partial roof leakage, coal dust clouds and instant floor swelling; the maximum floor heave of the roadway reaches 0.7 m.

## 2.3. Original support design

The roof and side of the roadway are covered by metal mesh attached to $\Phi 22 \times 2400$ mm rock bolts, and the bolt spacing is $900 \times 900$ mm. After the mesh and rock bolts are installed, cable bolts are installed at the top of the anchor net, with three anchor cables of $\Phi 22 \times 6300$ mm in the roof and two anchor cables of $\Phi 22 \times 4300$ mm on both sides of the roadway. The spacing between the anchor cables is $1800 \times 1800$ mm. Two $\Phi 22$ mm $\times 4300$ mm anchor cables are added to the two sides with a spacing of $1800 \times 1800$ mm. The roadway is supported by a 36U steel, 6.0 m support with a distance of 1200 mm, as shown in figure 3.

# 3. Countermeasures for the prevention and control of typical mines prone to rockburst

## 3.1. Countermeasures for support

This support is based on a strong–soft–strong structure with internal strong active support, soft structure pressure relief and anti-impact coordination. Considering the section of the roadway, the shape is rectangular: width × height = $5.8 \times 3.5$ m. The composite support is based on the combination of 'rock bolt and anchor cable support + hydraulic mop + soft structure anti-impact and energy absorption support'. $\Phi 22$ mm $\times L$ 2500 mm rock bolts with a spacing of $900 \times 800$ mm and $\Phi 18.9$ mm $\times L$ 5300 mm cable bolts

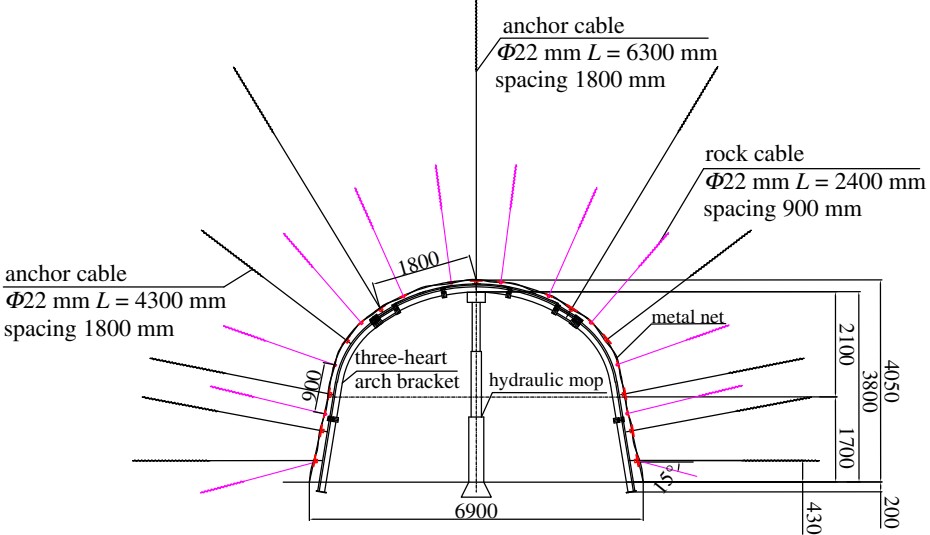

**Figure 3.** Original support design.

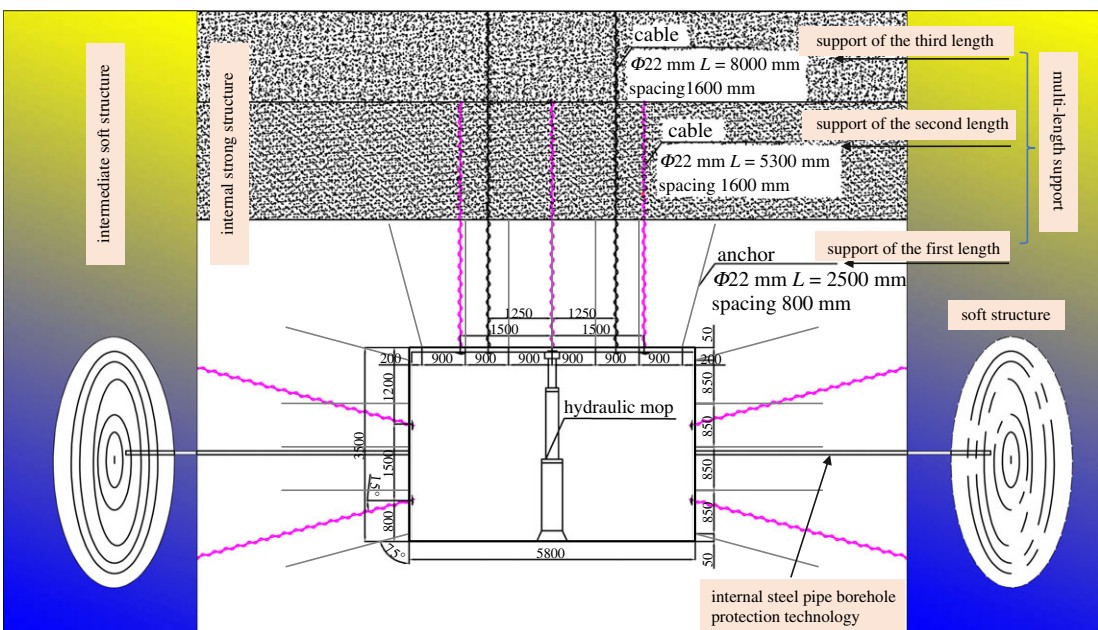

**Figure 4.** Design of supporting measures.

with a pattern of 2.5 × 1.6 m are installed along the roadway. Both sides of the roadway have $\Phi$22 mm × $L$ 2500 mm rock bolts with a pattern of 850 × 800 mm. There are 3.2 m long I-beams installed at the middle and bottom of the roadway sides. The rock bolts and cable bolts create a multi-length support system. In the centre of the roadway stands, a hydraulic mop is used to strengthen the support. The pressure relief holes are drilled in the sides of the roadway. For these holes, the borehole diameter is 110 mm, and the interval between the pressure relief holes is 3 m. Additionally, 10 m steel pipes are inserted into the boreholes to protect the support layer from damage. Drill pipe is used in the deep pressure relief holes to crack the coal and rock masses around the roadway. After cracking, the fractures in the coal and rock masses are connected to each other, and a soft structure on both sides of the roadway is formed, as shown in figure 4.

## 3.2. Monitoring method

To test the effect of roadway support and observe the deformation law of surrounding rock during roadway expansion and repair, the support design is scientifically validated [36,38,39]. Figure 5 shows

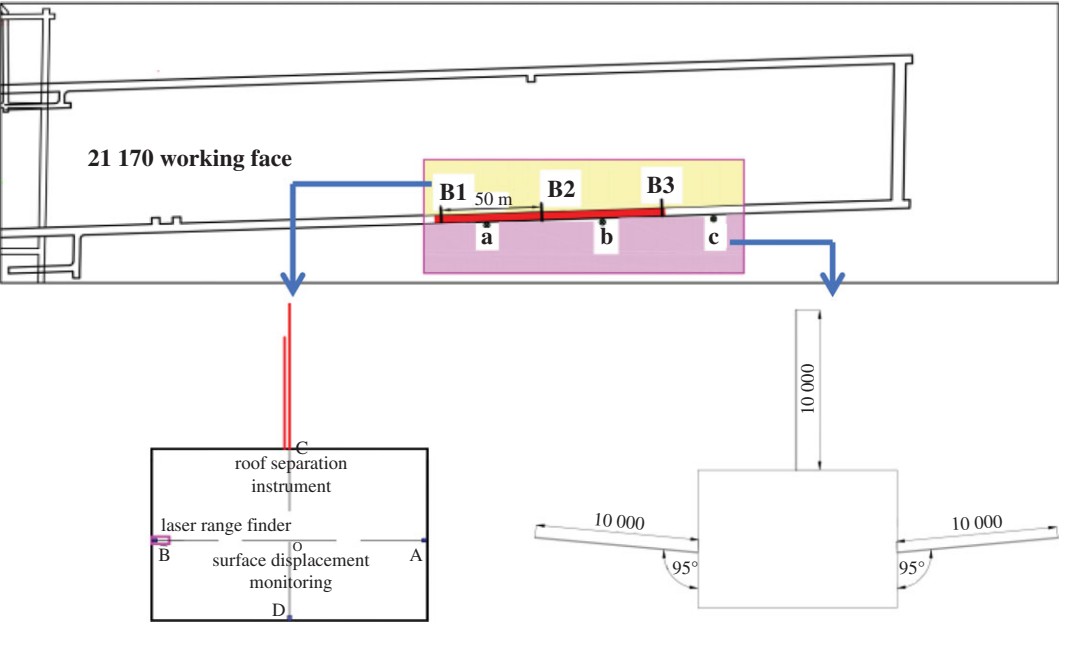

**Figure 5.** Monitoring locations.

the monitoring locations in the test areas. An M20-18-1000 bolt and OQ60 laser range finder are used for roadway surface displacement, and a 1 m long bolt is used to monitor the movement of the surrounding rock surface of the roadway. In this support design, the bolts located on the two sides and roof must be installed immediately after the completion of the roadway, and measurements of the initial state must be obtained. The bolts for monitoring the movement of the two sides of the roadway are installed centred between the two rows of bolts 1.5 m from the floor, and these two bolts must be kept horizontal; the small bolts that monitor the top and bottom of the roadway move closer to the centreline of the roadway over time. The roof layer separation failure uses a layer separation indicator, and the equipment for monitoring the displacement at each point in the surrounding rock of the roadway is a multi-point displacement meter, with a total of two measuring points of at 2.8 and 8.3 m from the floor. The installation time of the multi-point displacement meter represents the next phase of the roadway surface displacement monitoring. The installation must be carried out immediately after the construction of the roof anchor cable. The installation position of the multi-point displacement meter for monitoring the movement of the roof strata in the roadway is located at the centreline of the roadway and centred between the two rows of anchor cables. When installing the multi-point displacement meter, the drilling angle must be straight, the steel wire in the displacement meter must be fully tensioned and the initial reading must be zero. Figure 5a shows the test section of the 21 170 roadway. In the roadway, a measurement station is set every 50 m to observe the roadway surface displacement, roof separation, and bolt and anchor forces in the test roadway, and three monitoring points (B1, B2 and B3) are established. To monitor the fracture evolution, three points (a, b and c) are set to analyse the characteristics of the fracture evolution for the original support system and the coupled support system. The surrounding rock structure is observed by borehole images, which are obtained by a main machine and an auxiliary machine. The main machine is used to observe the development of fractures around the boreholes and display them on a screen. The auxiliary machine is composed of explosion-proof cameras to record the fractures. The depth of each of the roof and side boreholes is 10 m. All the obtained data are stored for later analysis.

# 4. Results and analysis

## 4.1. Evaluation of the 21 170 roadway under the original support system

The conditions of the 21 170 roadway under the original support system are shown in figure 6. The roadway roof had cracked and subsided greatly, and obvious cracks and fractures could be observed

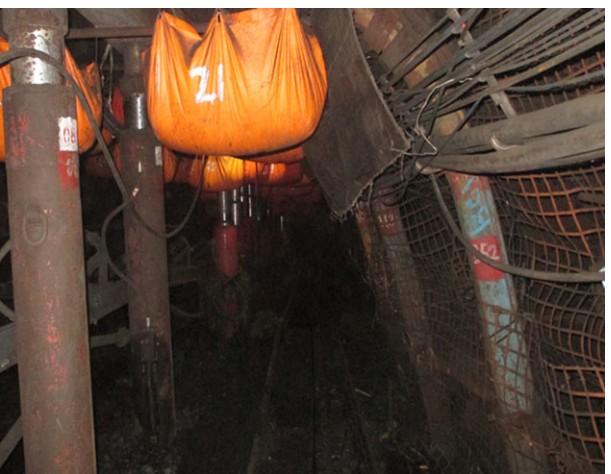

**Figure 6.** Photographs of the roadway under the original support system.

on a large scale. The deformation of both sides was large, and spalling was quite extensive. This accelerated the fracture evolution that led to rockburst in the roadway and to instability on both sides of the roadway, which intensified the roof deformation thus the rockburst events and ultimately leads to mine roadway destruction.

When the deep surrounding rock is damaged, the anchor point of a rock bolt is damaged, resulting in a rapid decrease in the anchoring force and ultimately rock bolt failure. A large burial depth and high vertical stress cause a high vertical load in the overburden rock at the sides of the roadway. Under the influence of high stress and pressure, the rock surrounding the roadway deformed, the cracks in the rock deformed and floor heave was the most prominent deformation mode. Note that the local position of the roadway support influences the stability of the overall structure of the surrounding rock. The large-diameter pressure relief boreholes on the sides of a roadway were used to prevent rockburst from greatly damaging the integrity of the coal body, which causes great damage to the overall integrity and stability of the rock surrounding the roadway. The rock bolts in the shallow coal body fractured the coal, and the fractures extended through the whole coal body. On both sides of the coal seam, fractures developed rapidly, and the coal seam became highly unstable.

Figure 7 shows the deformation measurements of the 21 170 roadway under the original support system. The monitoring points of roadway width, #1, #2 and #3 in figure 7a, are set at B1, B2 and B3 in figure 5a. The monitoring points of roadway height, #1 and #2 in figure 7b, are set at B1 and B2 in figure 5a. Monitoring point B3 shown in figure 5a was damaged, so the corresponding data are not analysed. The interval between the monitoring points in the original support system is 50 m. The roadway displacement changes within 60 days under the original support conditions. From day 0 to day 30, the roadway width at the monitoring points changes from 6050, 6010 and 6040 mm to 4700, 4613 and 4176 mm, respectively; the roadway height changes from 3820 and 3810 to 2860 and 3200 mm, respectively. From day 31 to day 60, the width of the roadway changes to 1570, 1620 and 1483 mm at monitoring points #1, #2 and #3; the height of the roadway changes to 1603 and 1592 mm at monitoring points #1 and #2. The displacements of the roof and side of the roadway increase sharply after the pressure relief boreholes are drilled, and finally, the roadway is completely destroyed. The destruction of the roadway sides causes the accelerated evolution of the fractures in the surrounding rock, which results in changes in the width and height of the roadway. The fractures reduce the roof integrity and cause the rapid deterioration of the two sides, and this dynamic deterioration further influences the overall stability of the roadway.

Figure 8 shows the damage zone in the rock surrounding the 21 170 roadway under the original support design. The maximum fracture depth of the roof reaches 5.03 m. Fractures are widely distributed to a depth of 3.2 m. The roof at 2.06–3.42 m is seriously damaged. The direct roof and the main roof are mudstones, which are thick, weathered and fractured. The annular fractures at 0.32 m in the roof block effectively transfer the rock bolt support force to the deep strata; consequently, the whole bolt support system is in an unstable state. Under strong dynamic loading, the rock strata are further separated, the surrounding rock structure of the roadway is instantaneously and completely voided, and the surrounding rock is rapidly squeezed into the open roadway space, resulting in sudden damage with characteristics of high energy, strong impact and short duration. At this time,

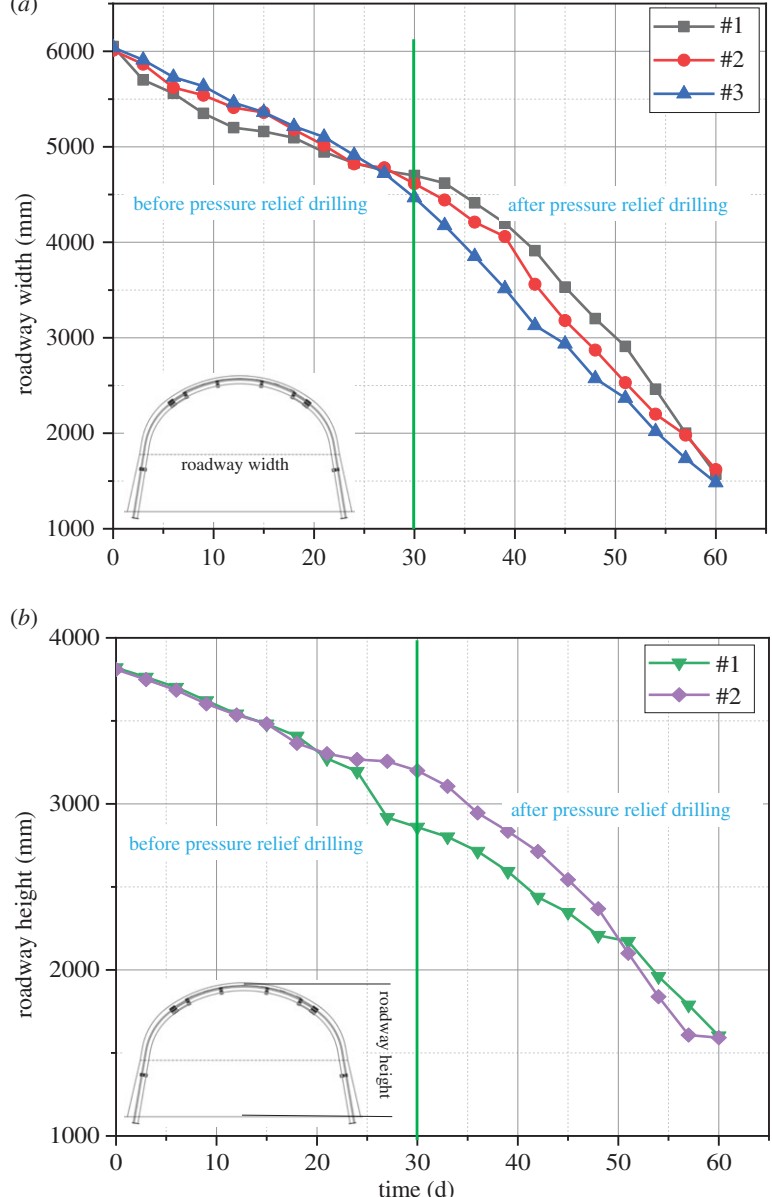

**Figure 7.** Diagram of roadway displacement under the original support system. (*a*) Roadway width. (*b*) Roadway height.

fractures are widely distributed within 4.53 m from the roadway. The roadway damage area is large, causing the roadway support to fail and the roadway to be seriously damaged.

The characteristics of roadway deformation and fracture distribution in the surrounding rock indicate that after a pressure relief borehole is drilled, the roof fractures that form are widely distributed within the range of the rock bolts and anchor cable supports. The superposition of the dynamic load causes the separation of the roof rock layers, resulting in the instability of the roadway roof. The deformation and instability of the roof increase the pressure on the roadway sides, causing the fractures in the roadway to penetrate and resulting in the destruction of the roadway sides.

## 4.2. Deformation characteristics of the 21 170 roadway under coupled support conditions

The test site is the 21 170 roadway failure location. Monitoring points #1, #2 and #3 in figure 7a are set at B1, B2 and B3 in figure 5a. According to the analysis of the effects in the test, the parameters designed for multi-length support technology are reasonable. Rock bolts (2.5 m) were used and combined with anchor cables (5.3 and 8.0 m) to form a multi-length support system. This system effectively controls the separation of the roof and restrains the deformation of the surrounding rock.

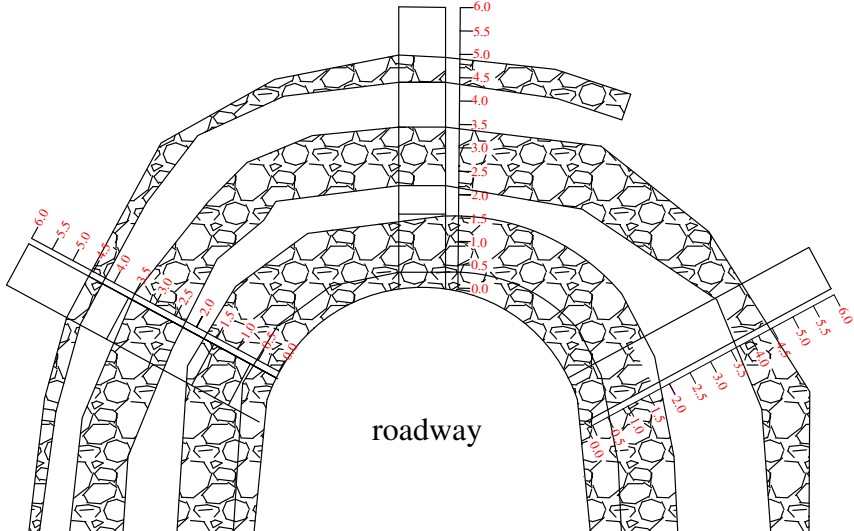

**Figure 8.** Damage zone under the original support system.

As shown in figure 9, the roof and side displacements recorded at each monitoring point are compared with those under the original support system: the displacement and roof subsidence of the roadway first increase sharply, then increase moderately and finally become stable. The difference between the side displacement and the roof subsidence is that the roof subsidence is smaller: the roof subsidence is between 300 and 500 mm, and the side displacement is between 500 and 600 mm. Before soft structure cracking, the roadway displacement and roof subsidence are 250 and 350 mm, respectively. After soft structure cracking, the roadway displacement and roof subsidence are 300 and 250 mm, respectively. A comparison of the deformations before and after soft structure cracking shows that the soft structures have little effect on the roadway stability. After 160 days, due to the impact of mining on the working face, the displacements of the two sides of the roadway increase.

The comparison and analysis of the recorded data indicate that the deformations under the coupled support conditions are smaller than those under the original support conditions. The influence of the soft structures on the roadway support is small. The multi-length support integrates the roadway roof and anchoring as a whole and closely combines the deep surrounding rock with the shallow surrounding rock. The coupled support system maintains the integrity of the roadway roof, reduces the tensile stress in the roadway roof and the damage to the roadway sides and absorbs the rockburst energy.

## 4.3. Soft structure

Soft structure cracking is caused by borehole drilling. The soft structure of the surrounding rock is drilled to form pressure relief boreholes, and the pressure release, absorption and transfer of the soft structure are realized by repeated borehole drilling. The soft structure transfers a considerable amount of stress and energy; impact waves attenuate through the softening area, which acts as a 'filter' that dissipates waves and absorbs energy. After soft structure cracking, there is an obvious stress reduction, which indicates that the soft structure plays a role in pressure relief, stress absorption and transfer. Borehole camera images are used to observe the soft structure. In the range of 10–15 m, the coal and rock masses are fractured. Microseismic monitoring is used to monitor the microseismic activity during the experiment. Figure 10 shows the microseismic energy monitoring of the roadway under the original support conditions and new support conditions, and the energy monitored by roadway microseismic monitoring is significantly reduced in new support conditions. The stress in the coal body is transferred or absorbed, which effectively reduces the roadway damage caused by the high stress and rockburst.

## 4.4. Evolution of surrounding rock fracture under coupled support conditions

The roof is composed of mudstone and sandstone. There are some very small fractures in the roadway mass within the range of the rock bolts, and no fractures are found outside the range of the rock bolts. As shown in figure 11, the maximum depth of roof fractures before the soft structure is formed is 2.59 m,

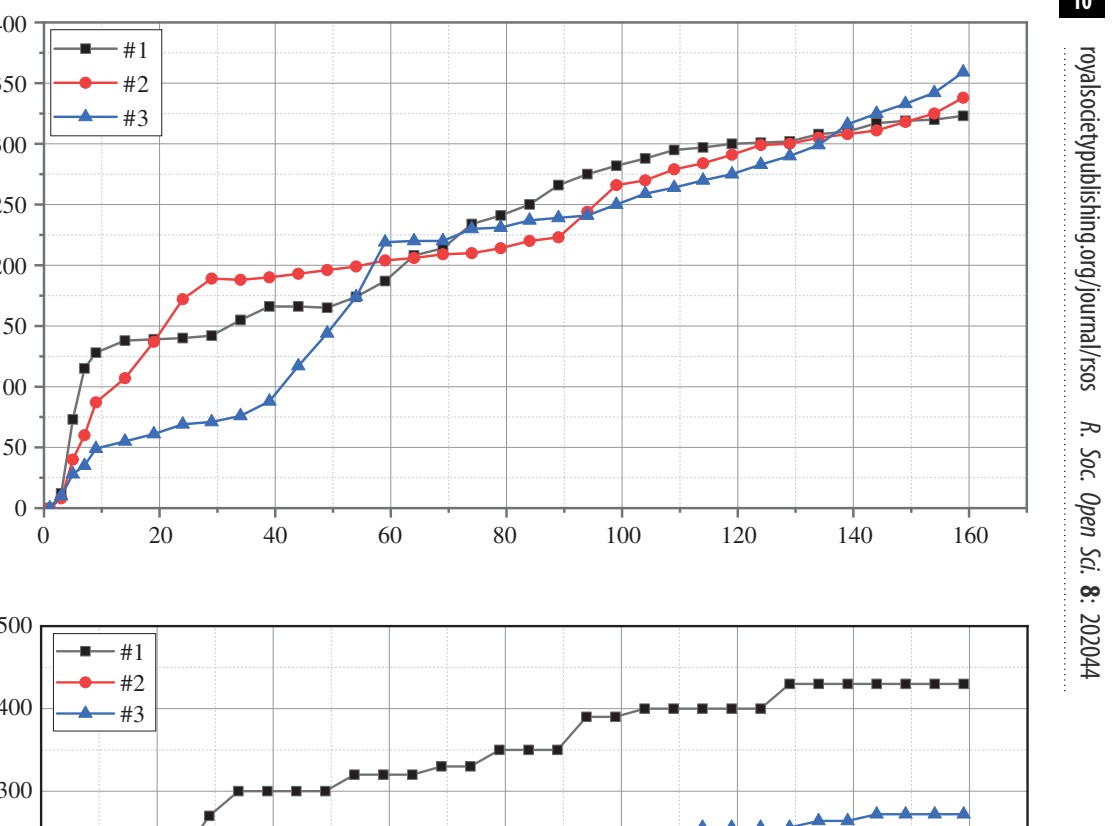

**Figure 9.** Displacements under coupled support conditions. (*a*) Change in roadway width. (*b*) Change in roadway width.

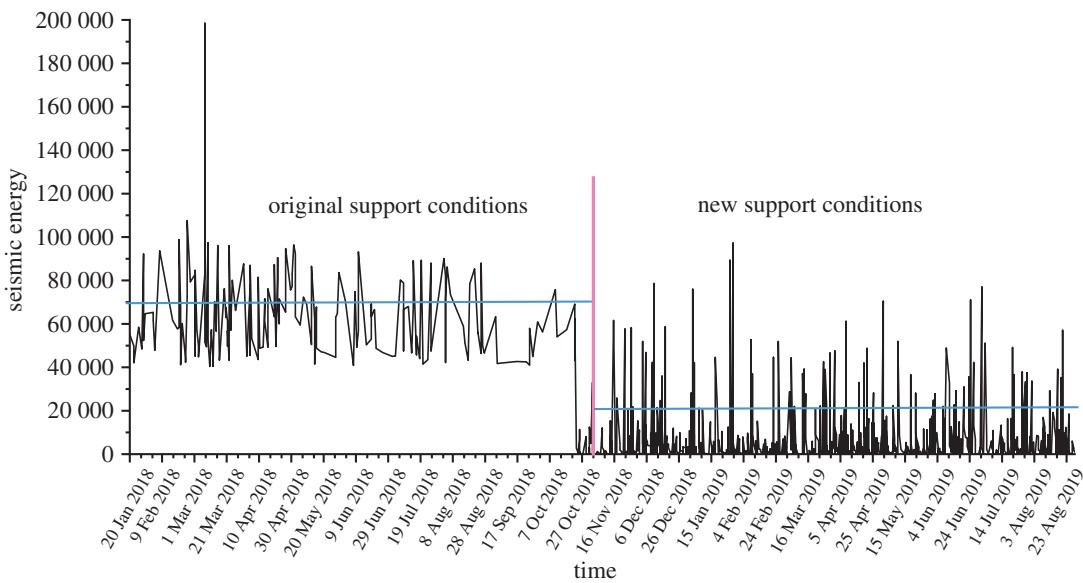

**Figure 10.** Microseismic energy monitoring of the roadway under the original support conditions and new support conditions.

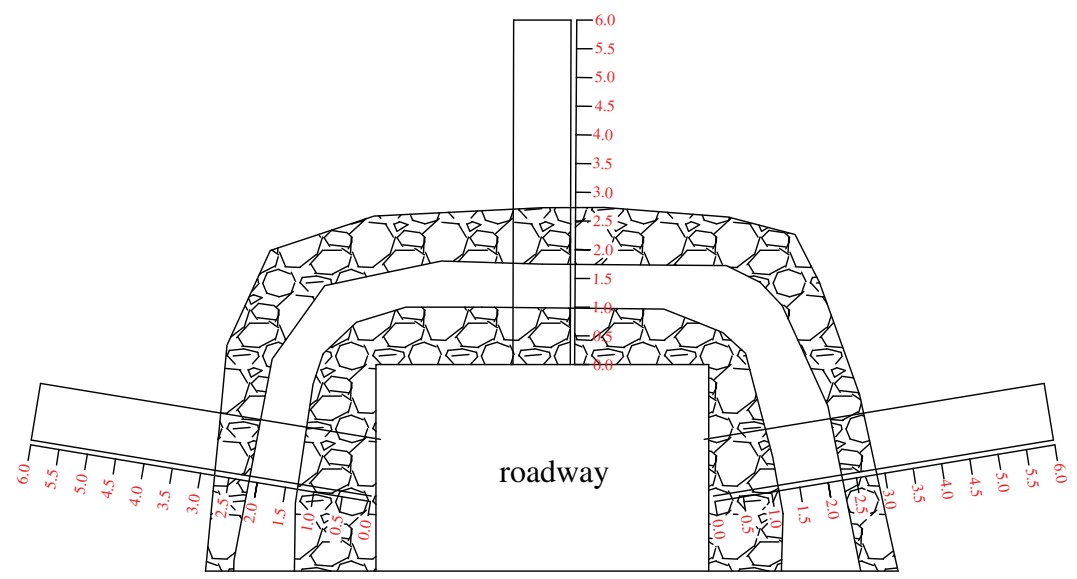

**Figure 11.** Damage zone under coupled support conditions.

compared with the maximum depth of 5.03 m under the original support conditions. The maximum failure depth of the roof under the coupled support system is reduced by 51% compared to that under the original support system. The maximum depth of the sidewall fractures under the coupled support system is 2.96 m, which is 4.56 m less than that under the original support conditions. The maximum failure depth of the roof under the coupled support system is reduced by 69% compared to that under the original support system. Coupled support effectively controls roadway roof fractures, achieving better support than the original support design. Using the technology of multi-length support and soft structure cracking, the fracture depth in the roof is obviously decreased, and the safety is significantly increased.

The maximum fracture depth is 2.48 m, and there are only very small fractures in the range of 0–0.29 m. Compared with the original support system, the coupled support system achieves thorough control over sidewall deformation. Under the conditions of coupled support, the roof deformation is effectively controlled. The pressure from the roof to the side clearly decreases, which restrains the evolution of minor fractures on the sides of the roadway and maintains the integrity of the roadway sides. The coupled support system greatly improves the conditions of the roadway from those achieved under the original support system.

## 5. Discussion

### 5.1. Model of roadway support measures

Figure 12*a* shows the strong–soft–strong structural model. The internal strong structure is the support structure in the rock surrounding the roadway; this roadway support layer is used to enhance roadway stability. The intermediate soft structure is the wave-absorbing area. The loose coal and rock mass formed by fracturing is used to absorb the energy generated by rockbursts. The external strong structure is a stable layer and is composed of an undisturbed original rock mass. Figure 12*b* shows the roadway deformation model. The borehole forms an elastic zone and a plastic zone. During the formation of the pressure relief zone around the borehole, the elastic energy stored in the coal body is partially released. The pressure relief zone of the borehole expands, which destroys the support structure, reduces the bearing capacity and causes the redistribution of stresses in the coal and rock masses.

The strong–soft–strong structure considers roadway pressure relief while maintaining the integrity of the roadway bearing structure to realize the theory of pressure relief and support for a rockburst occurring in a mine roadway. After pressure relief, the stability of the roadway decreases, the deformation rate increases and the support system fails. After repeated or excessive pressure relief, the coal and rock masses have no bearing capacity, and the roadway bearing system fails completely.

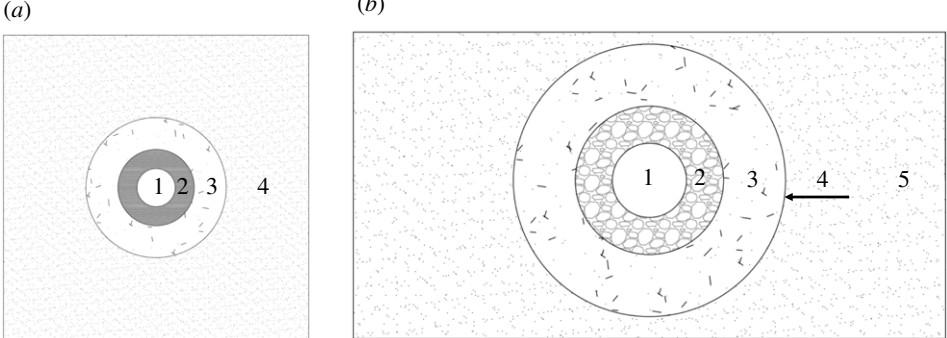

**Figure 12.** Model comparison. (*a*) Strong–soft–strong structural model. 1, roadway; 2, internal strong structure (6–12 m); 3, intermediate soft structure (8–10 m); 4, external strong structure (surrounding formation). (*b*) Roadway deformation model. 1, roadway; 2, broken zone; 3, plastic zone; 4, pressure relief zone boundary; 5, surrounding formation.

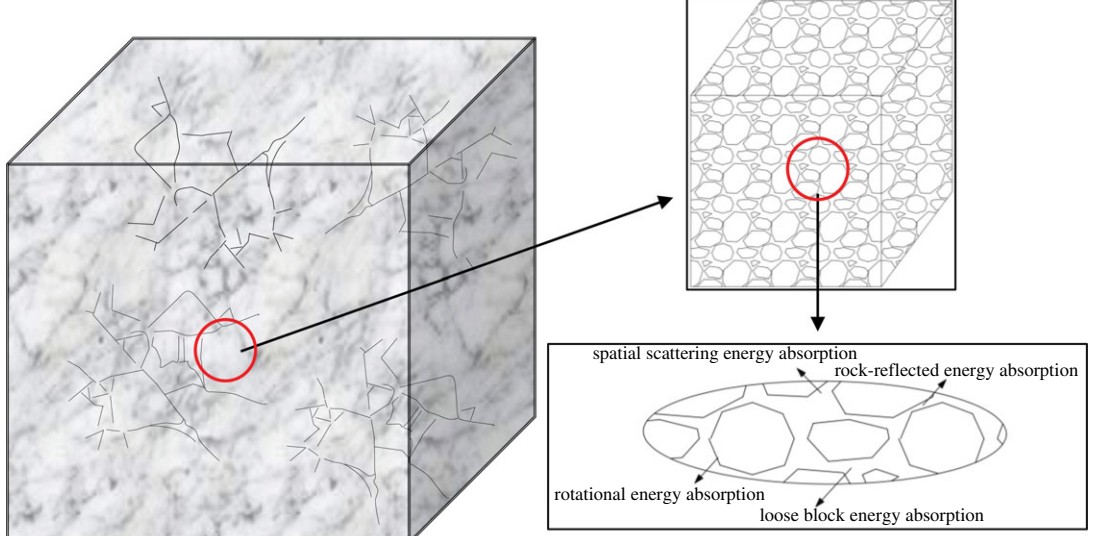

**Figure 13.** Soft structure.

## 5.2. Energy absorption of the soft structure

Loose coal and rock masses can provide shock isolation and energy absorption effects. The loose coal and rock masses formed by fracturing undergo a complex process of energy dissipation. As shown in figure 13, the soft structure is an artificially loose and fractured area, and its strength is basically reduced to the minimum. The energy absorption of the soft structure is mainly divided into three stages. In the static load stage, the soft structure undergoes a small deformation and absorbs energy. In the impact stage, the energy absorption of the soft structure is directly proportional to the relative density of the coal and rock masses, which is mainly reflected in the absorption of loose energy, rotation energy, spatial scattering energy and reflection energy. The shape and thickness of the soft structure determined by the cracks and particle sizes in the soft structure influence the attenuation and energy consumption of shock waves. In the transition stage, the rate of increase in energy absorption per unit volume decreases with increasing relative density. The increase in relative density improves the stiffness of porous materials, which enhances the energy translation performance of porous materials. Therefore, the soft structure attenuates and absorbs the impact stress waves and has a significant effect on the stability maintenance of the roadway in this deep high-stress environment.

## 5.3. Multi-length support technology

According to the mining operation and roof and floor characteristics of the Changcun coal mine, a three-dimensional calculation model is established. The height of the model is 112 m, the along-dip length of

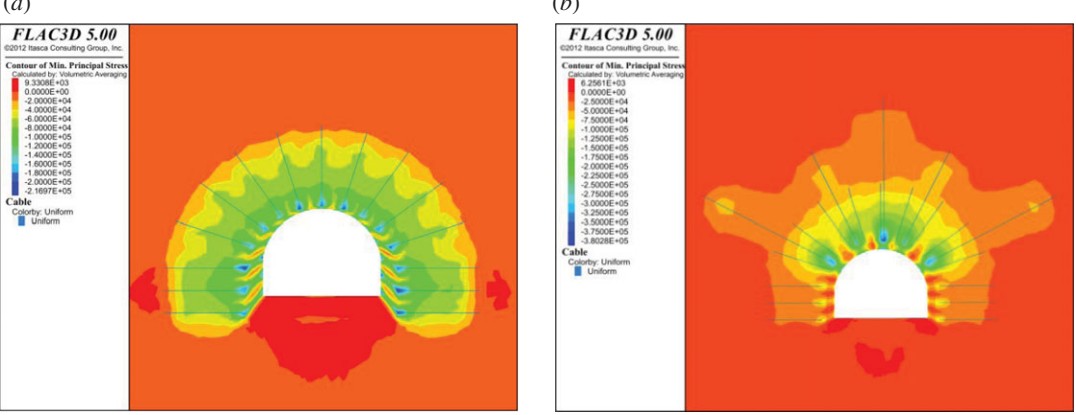

**Figure 14.** Distribution of rock bolt pre-tension stress field. (*a*) Rock bolt support. (*b*) Rock bolt with anchor cable support.

**Table 2.** Numerically calculated rock mechanics parameters.

| rock formation | lithology | density (N m$^{-3}$) | bulk modulus (GPa) | shear modulus (GPa) | adhesion (MPa) | friction angle (°) | tensile strength (MPa) |
|---|---|---|---|---|---|---|---|
| roof | overlying strata | 2530 | 8.0 | 7.6 | 2.0 | 32 | 1.29 |
| | siltstone | 2530 | 8.0 | 7.6 | 2.0 | 32 | 1.29 |
| | mudstone | 2220 | 12.2 | 12.0 | 2.0 | 31 | 0.60 |
| coal seam | coal | 1280 | 5.9 | 0.2 | 0.6 | 28 | 0.33 |
| bottom | fine sandstone | 2640 | 18.6 | 18.2 | 2.7 | 30 | 1.20 |
| | sandstone | 2410 | 14.4 | 6.5 | 2.0 | 32 | 1.0 |
| | underlying rock formation | 2410 | 14.4 | 6.5 | 2.0 | 32 | 1.0 |

the coal and rock strata is 120 m and the along-strike length is 180 m. The roadway is the actual width of the roadway under the condition of the original support. In the model, the origin of the ($X$, $Y$, $Z$) coordinate system is fixed, and the positive $Y$-direction is the heading direction. According to the *in situ* stress measurement results, the boundary stress of the initialization model is used to simulate the weight and horizontal stress of the overlying strata. The Mohr–Coulomb yield criterion is selected for the constitutive model, and the values of the coal and rock mechanics parameters are shown in table 2.

With the multi-length support technology, rock bolts are first installed in the shallow surrounding rock in the roof of the roadway. Then, short anchor cables are installed to control the coal and rock in the middle and lower parts of the roof to form a secondary reinforced anchoring bearing structure. Additionally, long anchor cables are installed to implement integral combined anchoring of the formed second-order anchor cable to the deep coal and rock masses in the roof, and the roof is anchored several times in sequence. The multi-length support structure thickness and strength and the combined anchoring effects in the roof successfully control the deformation of the rock surrounding the roof.

To understand the whole anchoring effect of the multi-length support technology on the surrounding rock, FLAC3D software is used to simulate the anchoring stress fields of ordinary bolt support and rock bolt with anchor cable support. As shown in figure 14, the compressive stress in the shallow surrounding rock is 0.14 MPa when using only bolt support, and the compressive stress in the surrounding rock is 0.22 MPa when the rock bolt and anchor cable support is used for multi-length support technology, which is an increase of 57.14%. The anchoring effect of the anchor cable improves the overall bearing capacity of the thick roof coal. The support structure composed of rock bolts and anchor cables optimizes the stress state in the surrounding rock, improves the stability of the roadway roof and is conducive to controlling the overall stability of the surrounding rock. Given the characteristics of the

coal seam and mudstone in the 21 170 roadway roof in the Changcun coal mine, the strata are very easily separated. The stress state of the coal and rock masses is optimized to effectively control the uncoordinated deformation between the soft interlayers in the top coal mass and improve the overall stability and self-supporting capacity of the top coal and rock masses.

## 5.4. Evolution mechanism of the roadway fractures

The integrity of the roof has an important influence on the stress in the sides of a mine roadway in which rockburst occurs. The fractures in the rock surrounding the 211 170 roadway under the coupled support conditions are obviously different from those under the original support conditions. Under the original support conditions, the fractures are distributed both within and outside the range of the rock bolts, and the roof separation is large. The maximum depth of the roof fractures is 6.56 m, and the maximum depth of the side fractures is 6.75 m. Under the original support conditions, roof separation and deformation occur, creating a large tensile stress. Thus, the stress in the roadway sides increases.

Under the coupled support conditions, the fractures are distributed in the shallow surrounding rock. Compared with those under the original support system, the fractures before and after soft structure cracking display a trend of decreasing degradation. The fracture depth is illustrated in figure 12. After the soft structure cracks, the fractures in the roadway do not increase significantly. Under the multi-length support system, the anchoring layer of the roof increases, which prevents the further evolution of the fractures and transmits the stress in the roof to the deep surrounding rock. The pressure of the roof on the sides of the roadway is reduced so that fractures do not further develop in the coal and rock masses. Under the coupled support conditions, the integrity of the roadway sides provides better support for the roof and reduces the deformation of the roof, which is important due to the relationship between the deformation of the roof and the sides of the roadway.

# 6. Conclusion

Research on the coordination of pressure relief and support measures is key to the maintenance of mine roadways in which rockburst has occurred. Both the pressure relief and the support of mine roadways in which rockburst has occurred must be considered. By studying improvement measures in a typical mine roadway in which rockburst has occurred, a pressure relief and support model of the roadway is established. The conclusions obtained are as follows:

(1) Compared with the original support design, the coupled support reduces roof convergence and rib convergence by 51% and 69%, respectively. After the soft structure cracks, fractures do not develop around the roadway. The coupled support reduces the fracture density in the shallow surrounding rock of the bolt anchorage zones. Generally, the coupled support design strengthens the supporting effect, and roadway control is very successful.

(2) The strong–soft–strong model has become an effective model in mine roadway support for preventing rockburst events. The active reinforcement measures for the internal strong structure and the energy absorption effect of the soft structure are studied, and the key technology to protect the internal strong structure, absorb energy in the soft structure and provide internal steel pipe borehole protection is proposed. The contradiction between the strong active support and soft structure pressure relief can prevent further rockburst events from occurring in mine roadways.

(3) Compared with the original support conditions of mine roadways in which rockburst has occurred, the coupled support conditions are obviously improved. The combined support of the rock bolts (anchor bolts) and steel belts controls the anchoring areas of the roof and prevent displacement of the roadway. The multi-length support strengthens the shallow surrounding rock and the deep surrounding rock support to prevent sinking of the roadway roof.

(4) This study is the first on the soft structure technology of repeated borehole drilling. The pressure relief effect of the roadway prevents damage caused by dynamic impact loads and provides soft structure pressure relief and impact resistance. The soft structure transfers a considerable amount of stress, absorbs impact energy and prevents serious damage to the roadway due to rockbursts.

Data accessibility. Data are included as electronic supplementary material. For additional information on the data, please contact the corresponding author.

Authors' contributions. Y.H. and M.G. participated in the data analysis and drafted the manuscript. D.X. and X.Y. collected the data and participated in the data analysis. All the authors have approved the manuscript for publication.

Competing interests. The authors declare no competing financial interests.

Funding. Funding was provided by the National Natural Science Foundation Project of China (grant no. 51564044) and the State Key Laboratory of Coal Resources and Safe Mining, China University of Mining and Technology (grant nos. SKLCRSM15X02 and SKLCRSM18KF004).

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
