## [Peer Review File · Royal Society Open Science]

Review History

RSOS-202044.R0 (Original submission)

Review form: Reviewer 1

Is the manuscript scientifically sound in its present form?

Yes

Are the interpretations and conclusions justified by the results?

No

Is the language acceptable?

No

Do you have any ethical concerns with this paper?

No

Have you any concerns about statistical analyses in this paper?

No

Recommendation?

Major revision is needed (please make suggestions in comments)

Comments to the Author(s)

Dear Author

The idea of stress release around the mining roadway is quite interesting, but the way of the results performance is very poor. Some terms are wrong or sounds strange (multiple coal explosions, drilling rig drills pressure-relief holes stress environment, rockburst roadway), especially while describing the support scheme (Fig. 4).

It is proposed, firstly to make a technical correction, and then verify the manuscript by a native speaker appropriate for the subject of the article.

Afterwards, the points need comments and explanations are:

1. You did not provide the detailed information about the monitoring. Where have you installed the gauges, what gauges, what techniques etc.?
2. You showed the results of convergence (?) measurements only. Why? And what about load on bolts?
3. What seismic activity were observed during your experiment? You say that presented approach helps in roadway protection in rockburst hazard areas
4. The interpretation of borehole walls pictures is not easy. The side camera is better (see references below) than front one to analyze fractures and separations in the boreholes. I suggest to summarize the results in the table or make a scheme of damage zone in the picture. Did you carry out the investigations in a one roadway cross-section only?
5. You perform the maps of stress after numerical modelling. It is 3D software, but you don't give any details about the model (size, boundary conditions, geomechanical parameters!) and the results are in 2D. It needs to be improved.
6. I'm surprised that you refer only to Chinese authors. There are many different papers abroad about borehole camera investigations and roadway monitoring. For example see, please:

Monitoring

- 1) Małkowski, P., Niedbalski, Z., Majcherczyk, T., Bednarek, Ł.: Underground monitoring as the best way of roadways support design validation in a long time period. *Mining of Mineral Deposits*, 2020, 14(3), pp. 1-14.
- 2) Niedbalski, Z., Małkowski, P., Majcherczyk, T.: Monitoring of stand-and-roof-bolting support: Design optimization. *Acta Geodynamica et Geomaterialia*, 2013, 10(2), pp. 215-226.
- 3) Majcherczyk, T., Małkowski, P., Niedbalski, Z.: Rock mass movements around development workings in various density of standing-and-roof-bolting support. *Journal of Coal Science and Engineering*, 2008, 14(3), pp. 356-360

Borehole camera

- 1) Małkowski, P., Niedbalski, Z., Majcherczyk, T.: Endoscopic method of rock mass quality evaluation - new experiences. *42nd U.S. Rock Mechanics - 2nd U.S.-Canada Rock Mechanics Symposium*, 2008.
- 2) Majcherczyk, T., Małkowski, P., Niedbalski, Z.: Describing quality of rocks around underground headings: Endoscopic observations of fractures. In: *Proceedings of the International Symposium of the International Society for Rock Mechanics, Eurock 2005*, 2005, pp. 355-360.

There are also imprecise statements in the manuscript, either generalized or requiring explanation. Most of them were marked in the enclosed manuscript (Appendix A).

Decision letter (RSOS-202044.R0)

Dear Dr Yongliang

The Editors assigned to your paper RSOS-202044 "Investigation of the evolution and control of fractures in surrounding rock under different pressure relief and support in a rockburst roadway" have now received comments from reviewers and would like you to revise the paper in accordance with the reviewer comments and any comments from the Editors. Please note this decision does not guarantee eventual acceptance.

Please submit your revised manuscript and required files (see below) no later than 21 days from today's (ie 18-Feb-2021) date. Note: the ScholarOne system will 'lock' if submission of the revision is attempted 21 or more days after the deadline. If you do not think you will be able to meet this deadline please contact the editorial office immediately.

on behalf of Professor Zach Agioutantis (Associate Editor) and R. Kerry Rowe (Subject Editor)
openscience@royalsociety.org

Reviewer comments to Author:

Reviewer: 1

Comments to the Author(s)

Dear Author

The idea of stress release around the mining roadway is quite interesting, but the way of the results performance is very poor. Some terms are wrong or sounds strange (multiple coal explosions, drilling rig drills pressure-relief holes stress environment, rockburst roadway), especially while describing the support scheme (Fig. 4).

It is proposed, firstly to make a technical correction, and then verify the manuscript by a native speaker appropriate for the subject of the article.

Afterwards, the points need comments and explanations are:

1. You did not provide the detailed information about the monitoring. Where have you installed the gauges, what gauges, what techniques etc.?
2. You showed the results of convergence (?) measurements only. Why? And what about load on bolts?
3. What seismic activity were observed during your experiment? You say that presented approach helps in roadway protection in rockburst hazard areas
4. The interpretation of borehole walls pictures is not easy. The side camera is better (see references below) than front one to analyze fractures and separations in the boreholes. I suggest to summarize the results in the table or make a scheme of damage zone in the picture. Did you carry out the investigations in a one roadway cross-section only?
5. You perform the maps of stress after numerical modelling. It is 3D software, but you don't give any details about the model (size, boundary conditions, geomechanical parameters!) and the results are in 2D. It needs to be improved.
6. I'm surprised that you refer only to Chinese authors. There are many different papers abroad about borehole camera investigations and roadway monitoring. For example see, please:

Monitoring

- 1) Małkowski, P., Niedbalski, Z., Majcherczyk, T., Bednarek, Ł.: Underground monitoring as the best way of roadways support design validation in a long time period. *Mining of Mineral Deposits*, 2020, 14(3), pp. 1-14.
- 2) Niedbalski, Z., Małkowski, P., Majcherczyk, T.: Monitoring of stand-and-roof-bolting support: Design optimization. *Acta Geodynamica et Geomaterialia*, 2013, 10(2), pp. 215-226.
- 3) Majcherczyk, T., Małkowski, P., Niedbalski, Z.: Rock mass movements around development workings in various density of standing-and-roof-bolting support. *Journal of Coal Science and Engineering*, 2008, 14(3), pp. 356-360

Borehole camera

- 1) Małkowski, P., Niedbalski, Z., Majcherczyk, T.: Endoscopic method of rock mass quality evaluation - new experiences. *42nd U.S. Rock Mechanics - 2nd U.S.-Canada Rock Mechanics Symposium*, 2008.
- 2) Majcherczyk, T., Małkowski, P., Niedbalski, Z.: Describing quality of rocks around underground headings: Endoscopic observations of fractures. In: *Proceedings of the International Symposium of the International Society for Rock Mechanics, Eurock 2005*, 2005, pp. 355-360.

There are also imprecise statements in the manuscript, either generalized or requiring explanation. Most of them were marked in the enclosed manuscript

===PREPARING YOUR MANUSCRIPT===

===PREPARING YOUR REVISION IN SCHOLARONE===

- If you are providing image files for potential cover images, please upload these at this step, and inform the editorial office you have done so. You must hold the copyright to any image provided.
- A copy of your point-by-point response to referees and Editors. This will expedite the preparation of your proof.

- Ensure that your data access statement meets the requirements at <https://royalsociety.org/journals/authors/author-guidelines/#data>. You should ensure that you cite the dataset in your reference list. If you have deposited data etc in the Dryad repository, please include both the 'For publication' link and 'For review' link at this stage.
- If you are requesting an article processing charge waiver, you must select the relevant waiver option (if requesting a discretionary waiver, the form should have been uploaded at Step 3 'File upload' above).
- If you have uploaded ESM files, please ensure you follow the guidance at <https://royalsociety.org/journals/authors/author-guidelines/#supplementary-material> to include a suitable title and informative caption. An example of appropriate titling and captioning may be found at https://figshare.com/articles/Table_S2_from_Is_there_a_trade-off_between_peak_performance_and_performance_breadth_across_temperatures_for_aerobic_scope_in_teleost_fishes_/3843624.

Author's Response to Decision Letter for (RSOS-202044.R0)

See Appendix B.

Decision letter (RSOS-202044.R1)

Dear Dr Yongliang,

It is a pleasure to accept your manuscript entitled "Investigation of the evolution and control of fractures in surrounding rock under different pressure relief and support measures" in its current form for publication in Royal Society Open Science. The comments of the reviewer(s) who reviewed your manuscript are included at the foot of this letter.

on behalf of Professor Zach Agioutantis (Associate Editor) and R. Kerry Rowe (Subject Editor)
openscience@royalsociety.org

Appendix A

Investigation of the evolution and control of fractures in surrounding rock under different pressure relief and support in a rockburst roadway

Yongliang He^{1,2}, Mingshi Gao^{1,2*}, Dong Xu^{1,2}, Xin Yu^{1,2}

1. School of Mines, China University of Mining and Technology, Xuzhou, Jiangsu 221116, China

2. State Key Laboratory of Coal Resource and Safe Mining, China University of Mining and Technology, Xuzhou, Jiangsu 221116, China

Correspondence should be addressed to Mingshi Gao; cumt_gms@cumt.edu.cn

Abstract: This paper studies fracture evolution in the surrounding rock under different pressure relief and support in Changcun coal mine, Henan Province, China. Fracture evolution in the surrounding rock was measured by borehole images obtained with a digital panoramic borehole camera, and the surface displacement of the rockburst roadway was measured using the intersection point method. Based on the coal and rock properties, original support method and impact tendency of the roadway coal and rock masses, a coupled method using pressure relief and support for a rockburst roadway is proposed, which resolves the contradiction between pressure relief and support of the roadway. According to field monitoring data, the active support technology of strengthening internal strong structures and the method of drilling repeated boreholes in soft structures effectively solves the problem of pressure relief and support in a rockburst roadway. Field measurements indicate that the overall effect on the roadway under the coupled method of pressure relief and support was more satisfactory than that under the original support. With the coupled support method, the surface displacement of the roadway was approximately 0.6 m, fractures were distributed only in soft structures and bolt anchorage areas, and the maximum depth of fractures was 2.95 m. In contrast, under the original supporting conditions, fractures were distributed throughout the whole roadway, and the maximum depth of fractures was 6.75 m. The reinforcement measures of multi-length support technology for internally strong structures, the protection measures of internal steel pipe boreholes, and the method of repeated borehole drilling in soft structures are proposed. The coupled roadway technology of "pressure relief + support" effectively maintains the stability of the rock surrounding the roadway and ensures the safety of the working face. The research results can provide a reference for damage prevention and support of rockburst roadways.

Keywords: rockburst; roadway support; fracture evolution; soft structure; multi-length support technology

1 Introduction

Rockburst is the sudden rupture and ejection of coal and/or rock masses from the surface of a roadway or working face, releasing a large amount of energy in an instant accompanied by loud noise and causing equipment and roadway damage; it may lead to gas and coal dust explosions and even casualties [1-5]. With the gradually increasing depth of coal mining, depths have exceeded 1000 m and reached 1500 m, and the intensity and frequency of rockbursts have increased significantly [6]. The number of rockburst coal mines increased from 142 in 2013 to 177 in 2017 in China [7]. Fig. 1 shows the number and distribution of rockburst mines operating in China to June 2019. From 2015 to 2020, partial rockburst accidents with serious casualties occurred in roadways in China, as shown in Table 1. According to statistics, 85% of rockburst accidents occur in roadways [8].

Fig. 1 Number and distribution of rockburst mines operating in China (to June 2019)

There are many studies on controlling roadway deformation and fracture propagation through pressure relief and support. The pressure-relief methods include deep hole pre-split blasting^[9-11], roof directional hydraulic fracturing [12, 13], coal seam blasting pressure relief [14-16], coal seam large-diameter pressure relief, coal seam high-pressure water injection [17], bottom blasting, floor cutting, etc. *↓ boreholes ?*

Table 1 Partial rockburst roadway accidents in China from 2015 to 2020

Date	Rockburst coal mine	Rockburst situation
May 26, 2015	Liaoning Fuxin Mining Group, Aiyou Coal Mine	Killed 4 coal miners
December 22, 2015	Henan Dayou Energy Co., Ltd, Gengcun Coal Mine	Roadway damage 160 m; killed 2 coal miners
July 22, 2016	Heilongjiang Longmei Group, Dongbaowei coal mine	Roadway damage 120 m; killed 2 coal miners
August 15, 2016	Shandong energy Feicheng Mining Group, Liangbaosi Energy Co., Ltd	Killed 2 coal miners
January 17, 2017	China Coal Group, Danshuigou Coal Industry Co., Ltd	Killed 10 coal miners
November 11, 2017	Liaoning Shenyang Coking Coal Co., Ltd, Hongyang No. 3 coal mine	Roadway damage 220 m; killed 10 coal miners
October 20, 2018	Shandong Energy Group, Longyun Coal Industry Co., Ltd	Roadway damage 100 m; killed 21 coal miners
June 9, 2019	Jilin Longjiabao Mining Co., Ltd	Roadway damage 220 m; killed 9 coal miners
August 2, 2019	Kailuan (Group) Co., Ltd, Tangshan Mining Branch	Killed 7 coal miners
February 22, 2020	Shandong Xinjulong Energy Co., Ltd	Killed 4 coal miners

Kang Hongpu et al. [18, 19] found that coal and rock masses form the main bodies of roadways bearing dynamic and static loads and that rock bolts improve the bearing capacity. The theory of high pre-stress and strong support is proposed to meet the initial support of the roadway and avoid roadway destruction. Ju Wenjun et al. [20, 21] suggested the equivalent section support principle on the internal stress arch structure and considered the characteristics of the full-section pressure appearance of the rockburst roadway. Gao Mingshi^[22, 23] proposed the strong-soft-strong structure theory of rockburst roadways and analysed the wave absorption and energy absorption characteristics, failure criterion and energy absorption effect. Pan Yishan^[24, 25] described the flexible-coupling fast-yielding support based on the pendulum wave theory, fast energy absorption and yielding support. Pan Junfeng^[26, 27] proposed multi-scale source control to prevent and control deep coal and rock dynamic disasters and studied the theory and technology of dynamic and static load sources to prevent and control rockbursts of deep mine roadways. LV Xiangfeng^[28] studied the propagation and attenuation of stress waves in roadways, obtained the energy and stress criterion of stress waves in the surrounding rock, and proposed a rigid-flexible coupled support system. Kang and Wu^[29, 30] assessed the response of rock bolts under dynamic and static loads in coal mines with impact tendency, and different series of rock bolts with high strength, elongation and impact toughness have been developed. He^[31-37] et al. developed constant-resistance-large-deformation (CRLD) bolts and established a dynamic CRLD

bolt model. An anchor cable with high constant resistance and large deformation produces
instantaneous deformation under dynamic loading; the cable has special mechanical properties to
maintain constant resistance. Pan Yishan et al. [38, 39] found that the failure of rock bolts was due to
low elongation and poor energy dissipation performance in rockburst roadways. In addition, a new
type of anti-impact energy-absorbing hydraulic support, "O"-type shed and U-shaped steel, has been
developed to avoid damage by the energy of several large impacts. Lv Xiangfeng [40] used the static
and dynamic load method to study the energy absorption characteristics of foam materials and
concluded that aluminium foam material is suitable for energy-absorbing support materials.

With the Changcun coal mine as the research site, this paper proposes a method for pressure
relief and support coordination. The characteristics of roadway deformation and the fracture evolution
in the surrounding rock are studied with a borehole camera to reveal the pressure relief and support
coordination mechanism. The coupled technology of pressure relief and support is proposed to realize
the safety and support effects on rockburst roadways. The new technology design, guaranteeing
sustainable mining production, provides some practical reference for rockburst roadways with similar
problems.

2 General introduction to Changcun coal mine

2.1 Geologic conditions

The depth

(a) Stratigraphy of the test roadway

(b) Layout of the test areas

Fig. 2 General situation of the Changcun coal mine

The Changcun coal mine is located in Changcun town, Yima city, Sanmenxia city, Henan Province, China. The Changcun coal mine is designed to adopt inclined shaft development. The direct roof of the roadway is sandstone with a thickness of 0.1 ~ 9.5 m. The sandstone ^[41] contains water, the lower part is loose and collapses easily, and the upper part is dense and hard. The old roof is mudstone ^[42] with a thickness of 35.5 ~ 44.5 m. The mudstone is thick ^{but} and weathers and breaks easily. The direct bottom is carbonaceous mudstone with a thickness of 0.5 ~ 5.7 m; it is grey-black and massive with a high carbon content and readily swells in contact with water. The old bottom is 27.2 ~ 32.8 m clay rock with argillaceous cementation, as shown in Fig. 2 (a). The ground elevation of the 21170 working face is + 497 ~ + 549 m, and the working face elevation is -161.080 ~ -210.709 m. As shown in Fig. 2 (b), the upper part of the 21170 working face is the 21150 working face that has been mined, and the lower part is the 21190 working face that has not been designed.

3.2 Rockburst conditions and roadway failure characteristics

The burial depth of the 21170 working face exceeds 700 m, and the vertical stress exceeds 17.5 MPa. Due to the extensive concentration of production, the excavation interference is large, and the mining intensity is gradually increasing, resulting in rockburst. During the mining and production processes, dozens of shock events have occurred, and the energy is mainly concentrated at approximately 10^4 J to 10^6 J. According to the assessment of coal rock impact tendency ^[43] in the Changcun coal mine, the coal seam impact tendency is weak; the roof has no impact tendency. However, from the multiple coal explosions and impacts that have occurred on the actual site, the coal body is relatively hard, which triggers great harm to the roadway support and disrupts the surrounding rock stability. After a rockburst occurs, it causes varying degrees of damage to the

1 roadway, such as damage to the roadway support, belt bulging and twisting, partial roof leakage,
 suspended coal dust, and instant floor swelling; the maximum floor heave of the roadway reaches 0.7
 5 m. Using the comprehensive evaluation method for rockbursts [44] and the analytical method of
 empirical analogy, the risk of rockburst in the mine is evaluated. The mining depth is greater than
 700 m in a solid coal area, and this area presents medium risk. *why?*

12 3.3 Original supporting design

The roof and side of the roadway are covered by metal mesh attached with $\Phi 22 \times 2400$ mm
 rock bolts, and the *bolt pattern* spacing between rows is 900×900 mm. After the ~~anchor~~ mesh and rock bolts are
 installed, *anchor cables are laid on top of the anchor net, with three anchor cables of $\Phi 22 \times 6300$ mm*
 *in the roof?*
 *on the roadway* and two anchor cables of $\Phi 22 \times 4300$ mm on both sides of the roadway. The row
 spacing between the anchor cables is 1800×1800 mm. Two $\Phi 22 \times 4300$ mm anchor cables are added
 to the two sides with a spacing of 1800×1800 mm. The roadway is supported by a 36U, 6.0 m three-
 centre arch support with a distance of 1200 mm, as shown in Fig. 3. *fully grouted?*
 *section?*

Fig. 3. Original supporting design

35 3 Countermeasures for prevention and control of typical rockburst mines

38 3.1 Countermeasures for support

41 This support is based on the strong-soft-strong structure with internal strong active support, soft

structure pressure relief and anti-impact coordination. Considering the section of the roadway, the
 shape is rectangular, and width \times height = 5.8 m \times 3.5 m. The composite support is based on the
 combination of "rock bolt and anchor cable active support + hydraulic ~~shed~~ ^{prop} support + soft structure
 anti-impact and energy absorption support". The roof uses Φ 22 \times L 2500 mm rock bolts with spacing
 of 900 mm \times 800 mm and Φ 18.9 \times L 5300 mm anchor cables with spacing of 2.5 m \times 1.6 m along the
 roadway. Both sides of the roadway use Φ 22 \times L 2500 mm rock bolts with spacing of 850 mm \times 800
 7 mm. There are 3.2 m-long I-beams in the middle and bottom of the roadway sides. The rock bolts and
 8 anchor cables are composed of multi-length supports. In the centre of the roadway stands a hydraulic
 ~~shed~~ ^{prop} to strengthen the support. A drilling rig drills pressure-relief holes in the sides of the roadway.
 The borehole diameter is 110 mm, and the interval between the pressure-relief holes is 3 m. At the
 same time, 10 m steel pipes are inserted into the boreholes to protect the internal strong structures
 from damage. The drill pipe in the deep pressure-relief borehole is used to crack the coal and rock
 masses around the roadway. After cracking, the fractures in the coal and rock masses are connected
 to each other, and a soft structure on both sides of the roadway is formed, as shown in Fig. 4.

Fig. 4. Design of supporting countermeasures

3.2 Monitoring method

Fig. 5 Monitoring locations

Fig. 5 shows the monitoring locations in the test areas. Fig. 5 (a) shows the test section of the 21170 roadway. In the roadway, a measurement ^{station} is set every 50 m to observe the roadway surface displacement, roof separation, and bolt and anchor forces in the test roadway, and three monitoring points B1, B2 and B3 are established. To monitor the fracture evolution, three points a, b, c are set to analyse the characteristics of the fracture evolution for the original support system and the coupled support system. The surrounding rock structure is observed by borehole images, which are obtained by a main machine and an auxiliary machine. The main machine is used to observe the development of fractures around the boreholes and display them on a screen. The auxiliary machine is composed of explosion-proof cameras to record the fractures. The depth of the roof and side boreholes is 10 m. All obtained data are stored. The show de

4 Results and analysis

4.1 Evaluation of 21170 roadway under the original supporting conditions

Fig. 6 Photographs of the roadway under the original supporting conditions

The conditions of the 21170 roadway under the original support system are shown in Fig. 6. As illustrated in Fig. 6 (a), the roadway roof cracked and subsided greatly, and obvious cracks and fractures could be observed on a large scale. The deformation of both sides was large, and spalling was quite serious. The roadway width ranged from 6.0 m to 1.5 m due to the dynamic and static loads of the overlying strata and the large-diameter pressure-relief boreholes. This accelerated the fracture evolution of the rockburst roadway and led to instability on both sides of the roadway, which intensified the roof deformation in a vicious circle of rockburst roadway destruction.

When the deep surrounding rock is damaged, the anchor point of a rock bolt is damaged, resulting in a rapid decrease in the anchoring force and finally rock bolt failure, as shown in Fig. 6 (b). A large burial depth and high vertical stress cause high vertical load pressure on the overburden rock at the sides of the roadway. Under the influence of high stress and strong pressure, the rock surrounding the roadway cracks and deforms, and floor heave is the most prominent deformation, as shown in Fig. 6 (c). Note only the local position of the roadway support and not the stability of the overall structure of the surrounding rock, as shown in Fig. 6 (d). Rockburst prevention requires that the large-diameter pressure-relief boreholes in the sides of the roadway greatly damage the integrity of the coal body, which causes great damage to the overall integrity and stability of the rock surrounding the roadway. The rock bolts in the shallow coal body are seriously fractured, and the fractures extend through the whole coal body. On both sides of the coal seam, fractures develop rapidly, and the coal seam is highly unstable.

red
it's
down

it's
to n
any

No,
not ob

rock bolts
are fractured

rock
mass
deforms
and
cracks

Fig. 7 Diagram of roadway width

Fig. 7 shows the deformation measurements of the 21170 roadway under the original supporting conditions. The interval between monitoring points of the original support system is 50 m. The roadway displacement changes within 60 days under the original support conditions. From day 0 to day 30, the roadway width at the monitoring points changes from 6050 mm, 6010 mm, and 6040 mm to 4700 mm, 4613 mm and 4176 mm, respectively; the roadway height changes from 3820 mm and 3810 mm to 2860 mm and 3200 mm, respectively. From day 31 to day 60, the width of the roadway changes to 1570 mm, 1620 mm and 1483 mm; the height of the roadway changes to 1603 mm and 1592 mm. The displacements of the roof and side of the roadway increase sharply after the pressure-relief boreholes are drilled, and finally, the roadway is completely destroyed. The destruction of the roadway sides causes the accelerated evolution of the fractures in the surrounding rock, which results in deformations of the width and height of the roadway. The fractures reduce the roof integrity and cause the rapid deterioration of the two sides, and the dynamic deterioration of the side-roof-side further influences the overall stability of the roadway.

Fig. 8 ^{shows} displays borehole camera images and fractures in the rock surrounding the 21170 roadway under the original supporting conditions. Monitoring points a, b, and c are set on the roof and sides of the test area. Roof-1, Roof-2 and Roof-3 are the positions set on the roof, and Side-1, Side-2 and

again - a layout
of monitoring

Side-3 are the side positions corresponding to the roof. Roof-1, Roof-2, Side-1 and Side-2 are
borehole camera images before the drilling of the pressure-relief borehole; Roof-3 and Side-3 are
borehole camera images after the installation of the pressure-relief borehole. As shown in Fig. 8, for
Roof-1, Roof-2, Side-1 and Side-2, the maximum fracture depth of the roof reaches 5.03 m before
the pressure-relief borehole is drilled. Fractures are widely distributed within the depth range of 3.2
8 m. The roof at 1.56~3.42 m is seriously damaged. The direct roof and the old roof are mudstones,
which are thick and easily weathered and broken. The annular fractures at 0.32 m in the roof block
the effective transfer of the rock bolt support force to the deep strata; consequently, the whole bolt
support system is in an unstable state. Under strong dynamic loading, the rock strata are further
separated, the surrounding rock structure of the roadway is instantaneously and completely void, and
the surrounding rock is rapidly squeezed into the free space, which is a kind of sudden damage with
high energy, strong impact and short duration. From Fig. 8, the maximum fracture depth of the
roadway side reaches 4.56 m, and fractures are seriously developed within the range of 1.52 m. After
pressure-relief drilling, the maximum fracture depth in the roof reaches 6.56 m, and those in the side
reach 6.75 m in Fig. 8, Roof-3 and Side-3. Fractures are widely distributed within the range of 4.53
18 m. The pressure-relief borehole greatly damages the support of the roadway.

The characteristics of roadway deformation and fracture distribution in the surrounding rock
indicate that after the pressure-relief borehole is drilled, the roof fractures are widely distributed in
the range of rock bolts and anchor cable supports. The superposition of the dynamic load causes the
separation of the roof rock layer, resulting in the instability of the roadway roof. The deformation and
instability of the roof increase the pressure on the roadway sides, causing the fractures in the roadway
to penetrate and resulting in the destruction of the roadway sides. The overall damage to the roadway
is shown in Fig. 8.

Fig. 8 Borehole images under the original support conditions

4.2 Deformation characteristics of 21170 roadway under coupled supporting conditions

is it necessary

Fig. 9. Photographs of the roadway under the new supporting conditions

where? wider?

what does it mean?

(a) Displacement of roadway side

only?

(b) Displacement of roadway roof

Fig. 10 Displacements under coupled support conditions

The test site is the 21170 roadway failure location. Fig. 9 shows the roadway photographs under
the coupled support system. According to the analysis of the effects in the test, the parameters
designed for multi-length support technology are reasonable. The roof uses 2.5 m rock bolts, and the
*were used (in)*
self-stability of the surrounding rock occurs through a high pre-tightening force. Combined with 5.3
8 m and 8.0 m anchor cables to form multi-length support, this system effectively controls the
9 separation of the roof and restrains the deformation of the surrounding rock.

As shown in Fig. 10, the roof and side displacements recorded at each monitoring point are
compared with those under the original supporting conditions: the displacement and roof subsidence
of the roadway first increase sharply, then increase moderately, and tend to become stable. The
difference is that the displacement and the roof subsidence are relatively small: the roof displacement
is between 30 and 50 mm, and the side displacement is between 500 and 600 mm. Before soft structure
cracking, the roadway displacement and roof subsidence are 250 mm and 35 mm, respectively. After
soft structure cracking, the roadway displacement and roof subsidence are 300 mm and 25 mm,
respectively. The comparison of deformations before and after soft structure cracking shows that soft
structures have little effect on roadway stability. After 160 days, due to the impact of mining on the
working face, the displacements of the two sides of the roadway increase.

The comparison and analysis of the data recorded before and after soft structure cracking
indicate that the deformations under the coupled support conditions are smaller than those under the
original support conditions. The influence of soft structures on roadway support is small. Fig. 10
shows the integrity of the roof and coal sidewall without any damage. The multi-length support has
*what? pre-tension?*
high pre-tightening and anchoring forces, which integrate the roadway roof and anchoring as a whole
and closely combines the deep surrounding rock with the shallow surrounding rock. Meanwhile, the
hydraulic lift and yielding pipes absorb the energy of rockbursts and inhibit the deformation of the
roadway under the superposition of dynamic and static loads. The coupled support system maintains
the integrity of the roadway roof, reduces the tensile stress in the roof and the damage on the sides
and absorbs the energy of rockbursts.
how do you know about it?
?

4.3 Soft structure

As shown in Fig. 11, the soft structure cracking is caused by repeated borehole drilling. The soft
structure of the surrounding rock is drilled by using pressure-relief boreholes, and the pressure release,
absorption and transfer of the soft structure are realized by repeated borehole drilling. The soft

structure transfers high stress and high energy, and the shock wave attenuates through the softening
area and plays the role of a "filter" in dissipating waves and absorbing energy. After soft structure
cracking, there is an obvious stress reduction, which indicates that the soft structure plays a role in
pressure relief, stress absorption and transfer. Borehole camera images are used to observe the soft
structure. In the range of 10~15 m, the coal and rock masses are broken, and the effect of soft structure
structure. In the range of 10~15 m, the coal and rock masses are broken, and the effect of soft structure
cracking is better. *it means?* *And?*

Fig. 11 Borehole camera images under a soft structure

4.4. Evolution of surrounding rock fracture under coupled supporting conditions

Fig. 12 Borehole images under coupled support conditions

To observe the fracture distribution under coupled supporting conditions, three roof and sidewall borehole camera monitoring points are established in the 21170 roadway before and after the soft

structure. Roof 1-1, Roof 1-2, S1-1 and S1-2 are the borehole camera images before the soft structure;
Roof 1-3 and S 1-3 are the borehole camera images after the soft structure. The positions of
monitoring points are shown in Fig. 5; the fracture distributions in the roof and sidewalls are presented
in Fig. 12. 2

As shown in Fig. 12, for Roof 1-1 and Roof 1-2, the roof is composed of mudstone and sandstone.
There are some tiny and minor fractures in the range of rock bolt supports, and no fractures are found
outside the range of rock bolt supports. The maximum depth of roof fractures before the soft structure
is 2.59 m, compared with the maximum depth of 5.03 m under the original support conditions. The
maximum failure depth of the roof under the original support system is reduced by 51%. As shown
in Fig. 12, for S 1-1 and S 1-2, the maximum depth of the sidewall fractures is 2.96 m, which is 4.56
16 m less than that under the original support conditions. The maximum failure depth of the roof under
17 the original support system is reduced by 69%. As shown in Fig. 12, Roof 1-3 is the fracture depth in
the surrounding rock of the roof after soft structure cracking. When soft structure cracking occurs,
borehole wall protection technology is used to protect the internal strong structure. The surrounding
rock fractures form only within the range of rock bolt supports, the maximum depth is 2.07 m, and
the fractures in the range of 0.43~0.65 m are serious. Compared with the situation before the soft
structure, the fractures in the roadway roof decrease significantly, and all the fractures are distributed
in the range of rock bolt supports. Coupled support effectively controls roadway roof fractures,
achieving better support than the original support design. Using the technology of multi-length
support and soft structure cracking, the fracture depth in the roof is obviously decreased, and the
safety is significantly increased.

in the rock mass
you mean the distance?
what is this?
ensure better roadway stability?
you see

Fig. 12 shows the fracture depth in the roof after the soft structure for S1-3. The maximum
fracture depth is 2.48 m, and there are only tiny fractures in the range of 0~0.29 m. Compared with
the original supporting system, the coupled support system achieves thorough control over sidewall
deformation. Under the conditions of coupled support, the roof deformation is effectively controlled.
The pressure from the roof to the side clearly decreases, which restrains the evolution of minor
fractures on the sides of the roadway and maintains the integrity of the roadway sides. Fig. 9 illustrates
that the surfaces of the roadway sides are smooth, and almost no coal is falling off. Compared with
the original support system, the coupled support system greatly improves the conditions of the
roadway.

5 Discussion

5.1 Model of roadway support countermeasures

(a) Strong-soft-strong structural model

1, roadway; 2, internal strong structure (6~12 m);
3, intermediate soft structure (8~10 m);
4, external strong structure (the surrounding formation).

(b) Borehole pressure-relief model

1, borehole; 2, broken zone; 3, plastic zone;
4, pressure-relief zone boundary;
5, surrounding formation.

Fig. 13 Model comparison

Fig. 13 (a) shows the strong-soft-strong structural model. The internal strong structure is the support structure in the rock surrounding the roadway, which is the roadway support layer and is used to enhance roadway stability. The intermediate soft structure is the wave-absorbing area. The loose coal and rock mass formed by fracturing is used to absorb the energy generated by rockbursts. The external strong structure is the stable layer and is composed of an undisturbed original rock mass. Fig. 13 (b) shows the pressure-relief model of boreholes. The borehole forms a broken zone and a plastic zone. During the formation of the pressure-relief zone around the borehole, the elastic energy stored in the coal body is partially released. The pressure-relief zone of the borehole expands, which destroys the support structure, reduces the bearing capacity and causes the redistribution of stresses in the coal and rock masses.

The strong-soft-strong structure considers roadway pressure relief while maintaining the integrity of the roadway bearing structure and realizes the theory of pressure relief and support for a rockburst roadway. The borehole pressure relief separates the pressure relief and support and ignores the effect of pressure relief and roadway support in preventing rockbursts. After pressure relief, the stability of the roadway decreases, the deformation rate increases, and the support system fails. After repeated or excessive pressure relief, the coal and rock masses lack bearing capacity, and the roadway bearing system fails completely.

5.2 Energy absorption of soft structure

Fig. 14 Soft structure

Loose coal and rock masses have shock isolation and energy absorption effects. The loose coal and rock masses formed by fracturing involve a complex process of energy dissipation. As shown in Fig. 14, the soft structure is an artificially loose and broken area, and its strength has been basically reduced to the minimum. The energy absorption of the soft structure is mainly divided into three stages. In the static load stage, the soft structure has small deformation and absorbs energy. In the impact stage, the energy absorption of the soft structure is directly proportional to the relative density of the coal and rock masses, which is mainly reflected in absorption of loose energy, rotation energy, spatial scattering energy and reflection energy. The shape of the soft structure by cracking and the particle size and thickness of the soft structure influence the attenuation and energy consumption of shock waves. In the transition stage, the rate of increase in energy absorption per unit volume decreases with increasing relative density. The increase in relative density improves the stiffness of porous materials, which enhances the energy absorption performance of porous materials. Therefore, the soft structure has attenuation and absorption effects on the impact stress waves and a significant effect on the stability maintenance of the roadway in the deep high-stress environment.

5.3 Multi-length support technology

The multi-length support technology first uses rock bolt supports in the shallow surrounding rock in the roof of the roadway. Then, short anchor cables are used to control the coal and rock in the middle and lower parts of the roof to form a secondary reinforced anchoring bearing structure. Additionally, long anchor cables are used to implement integral combined anchoring of the formed

second-order anchor cable to the deep coal and rock masses in the roof, and the roof is anchored several times in sequence. The multi-length support structure with thickness and strength and combined anchoring effects in the roof successfully control the deformation of the rock surrounding the roof.

(a) Rock bolt support

(b) Rock bolt + anchor cable support

Fig. 15 Distribution of rock bolt pre-tension stress field

To understand the whole anchoring effect of the multi-length support technology on the surrounding rock, the FLAC3D software is used to simulate the anchoring stress fields of ordinary bolt support and rock bolt + anchor cable support. As shown in Fig. 15, the compressive stress in the shallow surrounding rock is 0.14 MPa when using only bolt support, and the compressive stress in the surrounding rock is 0.22 MPa when the rock bolt + anchor cable is used for multi-length support technology, which is an increase of 57.14%. The anchoring effect of the anchor cable improves the overall bearing capacity of the thick roof coal. The support structure composed of rock bolts and anchor cables optimizes the stress state in the surrounding rock, improves the stability of the roadway roof and is conducive to controlling the overall stability of the surrounding rock. Given the characteristics of the coal seam and mudstone in the 21170 roadway roof in the Changcun coal mine, the strata are very easily separated. The stress state of the coal and rock masses is optimized to effectively control the uncoordinated deformation between the soft interlayers in the top coal mass and improve the overall stability and self-supporting capacity of the top coal and rock masses.

5.4 Evolution mechanism of roadway fractures

The integrity of the roof has an important influence on the stress in the sides of the rockburst roadway. Comparing Fig. 8 with Fig. 12 shows that the fractures in the rock surrounding the 21170 roadway under the coupled support conditions are obviously different from those under the original support conditions. Under the original support conditions, the fractures are distributed both within and outside of the range of rock bolt supports, and the roof separation is large. The maximum depth

of roof fractures is 6.56 m, and the maximum depth of side fractures is 6.75 m. Under the original
support conditions, roof separation and deformation occur, creating large tensile stress. Thus, the
stress in the roadway sides increases. Under greater tensile stress, the seriously damaged coal wall
triggers another round of roof deformation, forming a vicious circle.

Under the coupled supporting conditions, the fractures are distributed in the shallow surrounding
rock. Compared with those under the original supporting conditions, the fractures before and after
soft structure cracking display a trend of decreasing degradation. The fracture depth is shown in Fig.
12. After the soft structure cracks, the fractures in the roadway do not increase significantly. Under
the multi-length support system, the anchoring layer of the roof increases, which prevents the further
evolution of the fractures and transmits the stress in the roof to the deep surrounding rock. The
pressure of the roof on the sides of the roadway is reduced so that the fractures in the coal and rock
masses are not developed. The integrity of the roadway sides provides better support for the roof,
reduces the deformation of the roof, and forms a virtuous cycle between the roof and the sides of the
roadway. Under a rockburst, the soft structure absorbs the energy of the rockburst and maintains the
integrity of the roadway.

6 Conclusions

Research on the coordination of pressure relief and support is key to the maintenance of a
rockburst roadway. Both the pressure relief and the support of a rockburst roadway must be
considered. By studying improvement measures in a typical rockburst roadway, the pressure relief
and support model of the rockburst roadway are established. The conclusions obtained are as follows:

(1) Compared with the original support design, the coupled support reduces roof convergence
and rib convergence by 51% and 69%, respectively. After the soft structure cracks, the fractures in
the roadway do not develop. The coupled support reduces the fracture distribution in the shallow
surrounding rock of the bolt anchorage zones. Generally, the coupled supporting design strengthens
the supporting effect, and the roadway control is very successful.

(2) The strong-soft-strong model has become an effective model for rockburst roadway support.

The active reinforcement measures for the internal strong structure and the energy absorption effect
of the soft structure are studied, and the key technology to protect the internal strong structure, absorb
energy in the soft structure to and provide internal steel pipe borehole protection is proposed. The
contradiction between strong active support and soft structure pressure relief in the rockburst roadway

you did not measure any forces

is resolved.

(3) Compared with the original support conditions of the rockburst roadway, the support
situation of the roadway is obviously improved. The combined support of the rock bolts (anchor
cables) and steel belts controls the anchoring areas of the roof to prevent displacement of the roadway.
The multi-length support strengthens the shallow surrounding rock and the deep surrounding rock
support to prevent sinking of the roadway roof. The internal steel pipe borehole protection technology
protects the internal strong structure from being weakened. not described

(4) This study is the first to realize the soft structure technology of repeated borehole drilling.
The soft structure of the roadway provides a stress environment for roadway support. The pressure-
relief effect of the roadway avoids the damage caused by the impact dynamic load and forms the soft
structure pressure-relief and anti-impact features. The soft structure transfers high stress, absorbs
impact energy and avoids serious damage to the roadway caused by rockbursts.

Data accessibility. Data discussed in the present study were obtained from field observation. The data of field
observation results are described in detail in the 'Results and analysis (§4) of this manuscript. Additional data
required contact the corresponding author.

Authors' contributions. Yongliang He and Mingshi Gao participated and data analysis and drafted the manuscript.,
Dong Xu and Xin Yu collected the data and participated in data analysis. All authors approved the manuscript for
publication.

Competing interests. The authors declare no competing financial interests.

Funding. National Natural Science Foundation Project of China (grant no. 51564044) and the State Key Laboratory
of Coal Resources and Safe Mining, China University of Mining and Technology (grant nos. SKLCRSM15X02 and
SKLCRSM18KF004).

References

1. ZHOU J, LI X, MITRI H S. 2018 Evaluation method of rockburst: State-of-the-art literature review. *Tunnelling and*
*Underground Space Technology*. **81**,632-659. (doi:10.1016/j.tust.2018.08.029)
2. FENG XT, LIU J, CHEN B, et al. 2017 Monitoring, Warning, and Control of Rockburst in Deep Metal Mines.
*Engineering*. **3**, 538-545.(doi:10.1016/J.ENG.2017.04.013)
3. KENETI A, SAINSBURY BA.2018 Review of published rockburst events and their contributing factors. *Engineering*
*Geology*. **246**,361-373.(doi:10.1016/j.enggeo.2018.10.005)
4. LI T, CAI M F, CAI M. 2007 A review of mining-induced seismicity in China. *International Journal of Rock Mechanics*
*and Mining Sciences*. **44**, 1149-1171.(doi:10.1016/j.ijrmms.2007.06.002)
5. WANG K, DU F. 2020 Coal-gas compound dynamic disasters in China: A review. *Process Safety and Environmental*
*Protection*. **133**,1-17.(doi:10.1016/j.psep.2019.10.006)
6. RANJITH P G, ZHAO J, JU M, et al. 2017 Opportunities and Challenges in Deep Mining: A Brief Review. *Engineering*.
**3**, 546-51.(doi:10.1016/J.ENG.2017.04.024)
7. ZHU S, FENG Y, JIANG F. 2016 Determination of Abutment Pressure in Coal Mines with Extremely Thick Alluvium

Appendix B

Summary of Changes and Responses to Comments

We wish to thank the editor and reviewers for their insightful comments and suggestions regarding our paper, which considerably helped enhance the quality and readability of the manuscript. We have carefully considered all the suggestions from the editor and reviewers and modified the paper accordingly. Our point-by-point responses to the comments are presented below.

Comments to Reviewer #1

The idea of stress release around the mining roadway is quite interesting, but the way of the results performance is very poor. Some terms are wrong or sounds strange (multiple coal explosions, drilling rig drills pressure-relief holes stress environment, rockburst roadway), especially while describing the support scheme (Fig. 4).

It is proposed, firstly to make a technical correction, and then verify the manuscript by a native speaker appropriate for the subject of the article.

Response: Thank you for your comments. Following your insightful suggestion, we have carefully revised the paper. Moreover, a professional native English speaker has carefully reviewed the article. "Rock burst roadway" is a non-standard term. Consequently, we have changed "rock burst roadway" to "roadways in which rockburst has occurred" and "rockburst events in mine roadways". The whole article has been checked, and the inappropriate terms have been modified.

Comment 1: *You did not provide the detailed information about the monitoring. Where have you installed the gauges, what gauges, what techniques etc.?*

Response: To test the effect of roadway support and observe the deformation law of surrounding rock during roadway expansion and repair, the support design is scientifically validated. Fig. 5 shows the monitoring locations in the test areas. An M20-18-1000 bolt and OQ60 laser range finder are used for roadway surface displacement, and a 1 m long bolt is used to monitor the movement of the surrounding rock surface of the roadway. In this support design, the bolts located on the two sides and roof must be installed immediately after the completion of the roadway, and measurements of the initial state must be obtained. The bolts for monitoring the movement of the two sides of the roadway are installed centred between the two rows of bolts 1.5 m from the floor, and these two bolts must be kept horizontal; the small bolts that monitor the top and bottom of the roadway move closer to the centerline of the roadway over time. The roof layer separation failure uses a layer separation indicator, and the equipment for monitoring the displacement at each point in the surrounding rock of the roadway is a multi-point displacement meter, with a total of 2 measuring points of at 2.8 m and 8.3 m from the floor. The installation time of the multi-point displacement meter represents the next phase of the roadway surface displacement monitoring. The installation must be carried out immediately after the construction of the roof anchor cable. The installation position of the multi-point displacement meter for monitoring the movement of the roof strata in the roadway is located at the centreline of the roadway and centred between the two rows of anchor cables. When installing the multi-point displacement meter, the drilling angle must be straight, the steel wire in the displacement meter must be fully tensioned, and the initial reading must be zero. The force of the bolts is measured with an MCZ-300 anchor rod dynamometer. The dynamometer must be installed when the anchor rod is installed. The installation position of the dynamometer should be between the anchor rod and the anchor rod tray. When the force between the dynamometer and the tray is balanced, and the position of the dial should be easy to observe. Fig. 5 (a) shows the test section of the 21170 roadway. In the roadway, a measurement is performed every 50 m to observe the roadway surface

displacement, roof separation, and bolt and anchor forces, and three monitoring points (B1, B2 and B3) are established. To monitor the fracture evolution, three points (a, b, and c) are set to analyse the characteristics of the fracture evolution for the original support system and the coupled support system. The surrounding rock structure is observed by borehole images, which are obtained by a main machine and an auxiliary machine. The main machine is used to observe the development of fractures around the boreholes and display them on a screen. The auxiliary machine is composed of explosion-proof cameras to record the fractures. The depth of each of the roof and side boreholes is 10 m. All the obtained data are stored for later analysis.

Comment 2: *You showed the results of convergence (?) measurements only. Why? And what about load on bolts?*

Response: Roadway surface displacement or roadway surface convergence is an important index with which to measure the quality of roadway repair. The roadway surface displacement after repair is small, which can meet the requirements of the application, indicating that the roadway support scheme is effective. Because the case study in this paper is a repair roadway, the parameters such as bolt force are not recorded before repair, so a comparison cannot be made to reflect the difference before and after repair. Therefore, the bolt force and other parameters are not analysed in this manuscript. Because it will not affect the content of this paper, the discussion on the anchor force and bolt force has been removed from the discussion of the measurements and Fig. 5.

Comment 3: *What seismic activity were observed during your experiment? You say that presented approach helps in roadway protection in rockburst hazard areas*

Response: Microseismic monitoring is used to monitor the microseismic activity during the experiment. Fig. 10 shows the microseismic energy monitoring of the roadway under the original support conditions and new support conditions, and the energy monitored by roadway microseismic monitoring is significantly reduced in new support conditions. The stress in the coal body is transferred or absorbed, which effectively reduces the roadway damage caused by the high stress and rockburst.

Fig. 10 Microseismic energy monitoring of the roadway under the original support conditions and new support conditions

Comment 4: *The interpretation of borehole walls pictures is not easy. The side camera is better (see references below) than front one to analyze fractures and separations in the boreholes. I suggest to summarize the results in the table or make a scheme of damage zone in the picture. Did you carry out the investigations in a one roadway cross-section only?*

Response: Thank you for your comment. The borehole images are used to determine the damage zone. The main modifications can be found in the revised manuscript. A total of three tests were carried out in a roadway cross-section.

Comment 5: *You perform the maps of stress after numerical modelling. It is 3D software, but you don't give any details about the model (size, boundary conditions, geomechanical parameters!) and the results are in 2D. It needs to be improved.*

Response: Thank you for the question. We have added the details of the model. According to the mining operation and roof and floor characteristics of the Changcun coal mine, a three-dimensional calculation model is established. The height of the model is 112 m, the along-dip length of the coal and rock strata is 120 m, and the along-strike length is 180 m. The roadway is the actual width of the roadway under the condition of the original support. In the model, the origin of the (X, Y, Z) coordinate system is fixed, and the positive Y direction is the heading direction. According to the in situ stress measurement results, the boundary stress of the initialization model is used to simulate the weight and horizontal stress of the overlying strata. The Mohr-Coulomb yield criterion is selected for the constitutive model, and the values of the coal and rock mechanics parameters are shown in Table 2.

Table 2 Numerical calculation of rock mechanics parameters

Rock formation	Lithology	Density/ ($N \cdot m^{-3}$)	Bulk modulus/ (GPa)	Shear modulus/ (GPa)	Adhesion/ (MPa)	Friction angle/ ($^{\circ}$)	Tensile strength/ (MPa)
	Overlying strata	2530	8.0	7.6	2.0	32	1.29
Roof	Siltstone	2530	8.0	7.6	2.0	32	1.29
	Mudstone	2220	12.2	12.0	2.0	31	0.60
Coal seam	Coal	1280	5.9	0.2	0.6	28	0.33
Bottom	Fine sandstone	2640	18.6	18.2	2.7	30	1.20
	Sandstone	2410	14.4	6.5	2.0	32	1.0
	Underlying rock formation	2410	14.4	6.5	2.0	32	1.0

Comment 6: *I'm surprised that you refer only to Chinese authors. There are many different papers abroad about borehole camera investigations and roadway monitoring. For example see, please: Monitoring*

1) Małkowski, P., Niedbalski, Z., Majcherczyk, T., Bednarek, Ł.: Underground monitoring as the best way of roadways support design validation in a long time period. Mining of Mineral Deposits, 2020,

14(3), pp. 1–14.

2) Niedbalski, Z., Małkowski, P., Majcherczyk, T.: *Monitoring of stand-and-roof-bolting support: Design optimization. Acta Geodynamica et Geomaterialia*, 2013, 10(2), pp. 215–226.

3) Majcherczyk, T., Małkowski, P., Niedbalski, Z.: *Rock mass movements around development workings in various density of standing-and-roof-bolting support. Journal of Coal Science and Engineering*, 2008, 14(3), pp. 356–360

Borehole camera

1) Malkowski, P., Niedbalski, Z., Majcherczyk, T. *Endoscopic method of rock mass quality evaluation - new experiences. 42nd U.S. Rock Mechanics - 2nd U.S.-Canada Rock Mechanics Symposium*, 2008.

2) Majcherczyk, T., Małkowski, P., Niedbalski, Z.: *Describing quality of rocks around underground headings: Endoscopic observations of fractures. In: Proceedings of the International Symposium of the International Society for Rock Mechanics, Eurock 2005*, 2005, pp. 355–360.

Response: Thank you for your query. These references have been revised.

There are also imprecise statements in the manuscript, either generalized or requiring explanation. Most of them were marked in the enclosed manuscript

Response: Thank you for your query. The manuscript has been revised. The main modifications can be found in the revised manuscript.

We truly appreciate these helpful comments.

We have attempted to improve the manuscript and make all the necessary adjustments. The changes do not influence the content and framework of the paper. We have not explicitly listed the changes here but have marked them in green in the revised paper.

We appreciate the effort of the editor and reviewers and hope that the corrections are acceptable. Once again, we appreciate your comments and suggestions.